# Can we reliably estimate precipitation with high resolution during

# disastrously large floods?

3

2

1

- Jan Szturc<sup>1</sup>, Anna Jurczyk<sup>1</sup>, Katarzyna Ośródka<sup>1</sup>, Agnieszka Kurcz<sup>1</sup>, Magdalena Szaton<sup>1</sup>,
- Mariusz Figurski<sup>2</sup>, Robert Pyrc<sup>3</sup>

- Institute of Meteorology and Water Management National Research Institute, Centre of the Weather
- Forecasting Service, Warszawa, Poland
- <sup>2</sup> Gdansk University of Technology, Faculty of Civil and Environmental Engineering, Department of Geodesy,
- Gdańsk, Poland
- <sup>3</sup> Institute of Meteorology and Water Management National Research Institute, Hydrological and
- Meteorological Measurement and Observation Network Centre, Warszawa, Poland

Corresponding authors: Jan Szturc (jan.szturc@imgw.pl) and Anna Jurczyk (anna.jurczyk@imgw.pl)

16

20

22

24

- Abstract. A huge and dangerous flood occurred in September 2024 in the upper and middle Odra river basin, including mountainous areas in south-western Poland. The event provided an opportunity to investigate the feasibility of reliable estimation of high-resolution precipitation field, which is crucial for effective flood protection. Different measurement techniques were analysed: rain gauge data, weather radar-based, satellite-based, non-conventional (CML-based) and multi-source estimates. Apart from real-time and near real-time data, later available reanalyses based on satellite information (IMERG, PDIR-Now) and numerical mesoscale model simulations (ERA5, WRF) were also examined. Reference data used to verify the reliability of the different techniques for measurement and estimation of precipitation included observations from manual rain gauges and multi-source estimates from the RainGRS system developed at IMGW for daily and hourly accumulations, respectively. Statistical analyses and visual comparisons were carried out. Among the data available in real time the best results were found for rain gauge measurements, radar data adjusted to rain gauges, and RainGRS estimates. Fairly good reliability was achieved by non-conventional CML-based measurements. In terms of offline
- 1. Introduction
  - 1.1. Motivation
  - Precipitation is one of the most important meteorological parameters. In the case of extreme weather events, precise estimation of the precipitation field with high spatial resolution, preferably

reanalyses, mesoscale model simulations also demonstrated reasonably good agreement with reference

precipitation, while poorer results were obtained by all satellite-based estimates except the IMERG.

carried out in real-time, is of crucial importance for effective flood protection (Sokol et al., 2021; Velásquez et al., 2025), especially in mountainous regions. The accurate determination of precipitation amounts is also important for subsequent studies and expert opinions. In this context, the following question arises: Are we able to measure precipitation with sufficient reliability to carry out these tasks? The ability to estimate precipitation either in real time or in near real time (i.e. with a delay of up to several minutes, half an hour at most) is crucial, but data available afterwards for detailed analysis are also valuable.

Knowledge of the high-resolution spatial distribution of precipitation in real time provides the basis for generating forecasts with high resolution in time and space. Based on an extrapolation approach, nowcasting models (very short-range forecasting) generate such forecasts with very high precision but with a relatively short lead-time (Bojinski et al., 2023). This is particularly important when monitoring and forecasting severe convective phenomena (Fischer et al., 2024) for effective flood protection.

The main problem in analysing the accuracy of such forecasts is the lack of a reliable reference with a sufficiently high spatial and temporal resolution. Such a reference could be the most reliable measurements or re-analyses available offline. Manual rain gauge measurements, which are most often available in the form of daily accumulations, are usually used as a reference for other measurements and estimates (e.g. Hoffmann et al., 2016). However, rain gauges only provide point measurements, making spatial representation of precipitation highly dependent on network density. In the case of a sparse network and highly spatially variable precipitation, its accurate reconstruction becomes nearly impossible. Therefore, it is necessary to carry out various comparative analyses using all available measurement and estimation techniques to select optimal solutions (Hohmann et al., 2021; Loritz et al., 2021).

#### 1.2. State of the art

#### 1.2.1. High-resolution measurements of precipitation during extreme weather events

In the operational practice of the National Meteorological and Hydrological Services (NMHSs), the most commonly used rainfall measurement techniques are in-situ measurements made with various types of rain gauges, weather radar observations, and satellite-derived estimates. These measurements vary in spatial resolution, technical limitations, and sensitivity to various disturbing factors, and consequently, measurement errors have a completely different structure.

Rain gauges measure rainfall point-wise, i.e. only at their locations, and their reliability is affected by various factors related to meteorological conditions as well as to the failure rate and precision of the measurement, which is dependent on their design. This technique is considered the most accurate of those currently in use, but only in respect of the measurement location. Primarily, in the case of sparse rain gauge networks, point measurements do not provide reliable precipitation fields with sufficiently

high spatial resolution. One way to enhance the coverage of a given area with rain gauge measurements is to add data from personal weather stations (Garcia-Marti et al., 2023; Overeem et al., 2024).

Weather radars measure the spatial distribution of the precipitation field with a very high resolution of the order of 1 km, which depends on the distance from the radar site. However, radar data is sensitive to a wide variety of disturbances, such as the interaction of the radar beam with the terrain and objects on it, varying signal propagation conditions, interference with signals from other devices emitting microwave signals, e.g. RLAN (radio local area network) transmitters and many others. As a result, sophisticated quality control algorithms are necessary, although they are not completely effective (Méri et al., 2021; Ośródka and Szturc, 2022).

Operationally, the least reliable methods are those based on satellite imagery in the various spectral channels: microwave, which is the most technically challenging, as well as visible (VIS) and infrared (IR). Although satellite data are generally widely available, their reliability, except for microwave data, is relatively low, making them less commonly used in operational applications than rain gauge and radar data. In addition, their accuracy depends strongly on the season, time of day, and satellite location. A large number of satellite-based precipitation products have been designed using different spectral channels which are combined with other data, most commonly microwave active data from ground-based and satellite radars (e.g. GPM, Global Precipitation Measurement), microwave passive data from satellites in low polar orbits (e.g. MetOp of NOAA, National Oceanic and Atmospheric Administration), and mesoscale numerical model forecasts. This created the need for several comparative studies that were carried out in Europe, despite their much lower usefulness here (see, for example: Jiang et al., 2019; Navarro et al., 2020; Tapiador et al., 2020; Mahmoud et al., 2021; Peinó et al., 2025).

Additionally, precipitation data may come from devices not originally designed for meteorological measurements. The most common instance uses signal attenuation measurements on commercial microwave links (CML) from mobile phone networks (van der Valk et al., 2024; Olsson et al., 2025). These data require sophisticated algorithms to convert the measurements to precipitation, but they can provide many times more data than rain gauge networks. In Europe, attempts are being made to use these data in real time (Overeem et al., 2016; Nielsen et al., 2024; Graf et al., 2020; 2024; Olsson et al., 2025) taking advantage of the fact that networks of these kinds of links are very dense, especially in urbanised areas.

#### 1.2.2. Multi-source estimates

None of the measurement techniques described above demonstrates the ability to provide accurate precipitation estimation individually, but they are largely complementary. Considering that each has advantages and disadvantages, the idea is to combine data from different sources to improve the accuracy of rainfall estimation while maintaining high spatial resolution. Consequently, several merging methods have been developed to address the strengths and limitations of each measurement technique. They often include approaches based on conditional combinations of individual data (e.g., Sinclair and Pegram,

2005; Jurczyk et al., 2020b), the Kalman filter, and various versions of Kriging, such as Kriging with external drift (Sideris et al., 2014). Machine learning techniques, such as XGBoost (Mai et al., 2022; Putra et al., 2024), have been increasingly used for this purpose. Most often the merging process involves data from rain gauge and radar techniques (e.g., Goudenhoofdt and Delobbe, 2009; Ochoa-Rodriguez et al., 2019; Wijayarathne et al., 2020), and less often from the three combined techniques of rain gauge, radar and satellite (e.g., Jurczyk et al., 2020b; Yu et al., 2020; Putra et al., 2024). NOAA operationally provides the Multi-Radar Multi-Sensor (MRMS) quantitative precipitation estimates generated through integration of data from radar networks, surface and satellite observations, numerical weather prediction (NWP) models, and climatology (Zhang et al., 2016).

#### 1.2.3. Estimates based on numerical models

The surface or near-surface fields of precipitation simulated by numerical weather prediction (NWP) models are now frequently used for various purposes, including research of extreme precipitation events (Bližňák et al., 2022). Atmospheric reanalyses produced by NWP models with the assimilation of available historical observations can reconstruct past meteorological conditions. They provide physically consistent datasets of variables, including surface precipitation (Hersbach et al., 2020). The current NWP models are able to simulate intense precipitation, but the agreement with rain gauge observations is still not high in terms of spatial and temporal representation of precipitation (Bližňák et al., 2019).

For the characterisation of precipitation patterns, it is possible to use precipitation simulations obtained from NWP models, such as the publicly available ERA5 of ECMWF reanalyses (e.g., Subba et al., 2024). Other high-resolution mesoscale models with open-access software, such as WRF (Weather Research and Forecasting) of NCAR (Tanessong et al., 2017; Skamarock et al., 2019), can also be used. A significant upside to using such a solution, even in areas with dense in situ measurement networks, is the easy access to the data and their convenient processing.

## 1.2.4. Problems in the verification of precipitation measurements

Although several methods for verifying precipitation data have been developed over the years (e.g., Rodwell et al., 2011; Szturc et al., 2022), this issue is still challenging (Skok, 2022; Zhang et al., 2025). A fundamental problem in precipitation measurements is the considerable difficulty deriving information about precipitation on the ground surface, the so-called ground truth. Therefore, empirical verification of different measurement or estimation techniques is generally carried out indirectly through their intercomparison during field experiments. This process often involves a somewhat arbitrary selection of the most reliable measurement data or estimates based on the experience of the researchers. Rain gauges, especially manual ones, are believed to provide direct and relatively accurate data from point rainfall measurements. Thus, they are often considered the ground truth source for verifying other, mostly grid-based rainfall products, such as radar and satellite-based, multi-source, or NWP model

reanalyses (e.g., Militino et al., 2018). In a very sparse network of manual rain gauges, telemetric rain gauges can be used for this purpose, but only after advanced quality control.

The problem of precipitation data verification is much more difficult in mountainous areas due to the more significant spatial variability of precipitation distribution, which is associated with complex terrain (Ouyang et al., 2021). This aspect should also be kept in mind when verifying different types of measurements (Merino et al., 2021).

Furthermore, comparing the average precipitation over a grid area to a specific point value introduces some uncertainty, particularly during heavy rain (Ensor and Robeson, 2008). An analysis of findings by Sun et al. (2018), Herrera et al. (2019), and others shows that, due to the high spatial variability of precipitation, it is not possible to establish a single universal error value when comparing point and grid data. The level of the uncertainty varies depending on the nature of the precipitation. For widespread (large-scale) precipitation, the uncertainty typically ranges from about 10% to 15%. However, for intense, convective extreme precipitation, this uncertainty can rise to approximately 15% to 25% (Schellart et al., 2017; Henn et al., 2018; Tarek et al., 2021). Special care should be taken when analysing local precipitation maxima using gridded data, as noted by Sun et al. (2018) and others, who point out that these data may smooth out extreme values compared to point measurements.

## 1.3. Objectives and structure of the paper

The main objective of this work is to examine the real possibilities of precise estimation of a precipitation field with a high spatial resolution of about 1 km and a high temporal resolution of at least 10 min, or one hour during intense precipitation events that cause floods in upper Odra River basin area in September 2024. All available real-time and offline measurements and estimates were verified to determine their applicability and to quantify their reliability.

The paper is organised as follows: after an introductory Section 1 outlining the issues of precipitation measurement and the various techniques used, Section 2 briefly describes the 2024 flood event and the area affected. Section 3 details the precipitation data used in this work, both available in real time and with a delay for a longer period. Section 4 presents the results of the statistical verification of the data obtained by the different techniques and outcomes of the comparative analyses. Section 5 provides conclusions drawn from evaluating reliability of the investigated measurements and estimates.

#### 2. Flood in Poland in the Odra river basin in 2024

#### 2.1. Characteristics of the flooded area

The Odra (or Oder) is the second largest river in Poland. It forms part of the central European drainage network. The river starts in the Sudety Mountains in the Czech Republic and flows north, mainly through Polish territory, to the Baltic Sea. The river's total length is 855 km, and the maximum elevation in its basin is 1,602 m above sea level in the Sudety (Mount Śnieżka). After the Carpathian

Mountains, the Sudety have Poland's highest annual precipitation accumulation. At the same time, the area is characterised by high precipitation variability due to the complex orography, the natural increase in precipitation intensity with altitude, and the occurrence of precipitation shadows in the lower parts of the mountains and valleys.

Figure 1: The area of the upper and middle Odra river basin in Poland.

The rivers draining the Sudety Mountains and its foothills are prone to dangerous floods that can occur after high precipitation. The Odra River basin is characterised by numerous left-bank short tributaries draining rainwater from the mountains. Moreover, in the case of the Kłodzko Valley, there is a concentric system of river networks that favours the occurrence and dynamic of flood phenomena (e.g., Szalińska et al., 2014; Ligenza et al., 2021).

Rain-induced floods in the Odra river basin are usually associated with low-pressure frontal centres that reach Poland and cause prolonged and intense precipitation in southern of the country. In Poland, catastrophic rainfall floods occur most frequently just in the upper and middle Odra basin, with an area of approximately 44,000 km<sup>2</sup> (Fig. 1), on average every 10-15 years. The last ones were recorded in 1997, 2010, and 2024, the latter of which was investigated in this study.

The literature on analysing these floods is extensive, generally in Polish, but comprehensive English-language scientific studies can also be found. They address the subject from very different perspectives. Some studies cover a wider area than the Odra basin, e.g. the whole of Poland (e.g., Kundzewicz, 2014), central and eastern Europe (Bissolli et al., 2011), or the whole of central Europe (Mudelsee et al., 2004; Kimutai et al., 2024). Others describe and analyse in detail the course of floods (precipitation and river flows) in specific basins, e.g. the Odra River in Poland (Szalińska et al., 2014) or the Nysa Kłodzka River (Perz et al., 2023), which is an important tributary of the Odra River. Research suggests that climate change affects the frequency and severity of floods, leading to an increased risk of flooding (e.g. Kundzewicz et al., 2023). Detailed statistical analyses of rainfall during floods have also been carried out (e.g. Mikolajewski et al., 2025).

The above studies indicate that the upper Odra River basin is highly vulnerable to flooding caused by intense precipitation in the mountainous part of the basin. This is also influenced by the shape of the river network, which favours the cumulation of floods from individual tributaries. The flood risk there occurs almost annually during the summer.

# 2.2. Description of the flood

On 12-15 September 2024, the upper and middle Odra River basin and part of the upper Vistula River basin experienced rainfall that significantly changed the hydrological situation. From 12 September 2024, intense rainfall began to appear in western Poland, with accumulations of up to 60 mm in 12 hours recorded in the Eastern Sudety Mountains. The highest rainfall intensity occurred on consecutive days: from 13 September 2024 in the morning to 15 September 2024, before noon. The precipitation was associated with a low-pressure system named Boris by the national meteorological services of southern and central Europe.

# Figure 2: Field of precipitation accumulation during the flood of 13-16 September 2024 (four days) for the upper and middle Odra River basin in Poland, obtained from the multi-source RainGRS Clim estimates.

At many locations, the daily precipitation accumulation in this period exceeded 200 mm, and its territorial range covered mainly the Eastern Sudety Mountains. Four-day precipitation accumulation reached values above 400 mm, with the highest in the Jeseníky and Śnieżnik Mountains. They might have exceeded even 550 mm, as indicated by reanalyses RainGRS Clim (Jurczyk et al., 2023) based on estimates from the RainGRS system adjusted to observations from manual rain gauges (Fig. 2). Apart from intense, widespread precipitation, numerous thunderstorms and several associated tornadoes were recorded during these days. On 16 September, rainfall began to diminish; mainly light to moderate precipitation was observed, and in the following days, the weather in Poland was influenced by a high-pressure system, with the advection of warm and dry air of continental origin.

The consequence of the intensive rainfall was runoff of rainwater, high and extreme water levels in rivers, and flooding. The flood wave moved down the Odra River and its tributaries, causing numerous exceedances of warning and alarm levels.

# 3. Data used for the flood monitoring and analyses

### 3.1. The data used

In the frame of this study, the input data used to retrieve the precipitation field (Table 1) are divided into two groups in terms of the delay in their availability: (i) in real time and near real time, (ii) not in real time (with a delay of more than 30 min). Among the latter, data from manual rain gauges (GAU Manual), characterised by the highest reliability based on knowledge of measurement techniques and experience, were selected as reference data. All other precipitation products are verified by quantitative comparison with them.

Table 1. High-resolution techniques for measurement and estimation of the precipitation field.

| Abbreviation     | Description                                        | Temporal resolution | Spatial resolution  | Timeliness          |
|------------------|----------------------------------------------------|---------------------|---------------------|---------------------|
|                  | Reference da                                       | ata                 |                     |                     |
| GAU Manual       | Data from manual rain gauges (Hellmann's type)     | 24 h                | Point wise          | 2 months            |
|                  | Data available in 1                                | real time           | 1                   | 1                   |
| GAU              | Interpolated data from telemetric rain gauges      | 10 min              | 1.0 km              | 6 min               |
| RAD              | Weather radar data from POLRAD and                 | 5/10 min            | 0.5/1.0 km          | 4 min               |
|                  | neighbouring countries                             |                     |                     |                     |
| RAD Adj          | Weather radar data from POLRAD and                 | 5/10 min            | 0.5/1.0 km          | 7 min               |
|                  | neighbouring countries adjusted to telemetric rain |                     |                     |                     |
|                  | gauge data                                         |                     |                     |                     |
| SAT              | Satellite-based precipitation – combination of     | 5/10 min            | Roughly 3.5 km x    | 4 min               |
|                  | EUMETSAT NWC SAF products                          |                     | 6.0 km*             |                     |
| H61B             | Satellite-based precipitation – MW-IR combination  | 1, 24 h             | Roughly 3.5 km x    | 5-10 min            |
|                  | (EUMETSAT H SAF product)                           |                     | 6.0 km*             |                     |
| CML              | Interpolated estimates based on signal attenuation | 15 min              | 1.0 km              | Tests in progress   |
|                  | in commercial microwave links                      |                     |                     | (currently offline) |
| GRS              | Multi-source estimates from RainGRS system         | 10 min              | 1.0 km              | 7 min               |
|                  | Data available not in rea                          | l time (offlin      | e)                  | I                   |
| IMERG            | Satellite-based precipitation estimates of NASA,   | 30 min              | Roughly 7 km x 11   | About 4 months      |
|                  | final analyses (IMERG Final)                       |                     | km* (0.1° x 0.1°)   |                     |
| PDIR-Now         | Satellite-based precipitation estimates of         | 1 h                 | Roughly 2.8 km x    | 30-60 min           |
|                  | University of California, Irvine                   |                     | 4.5 km* (0.04° x    |                     |
|                  |                                                    |                     | 0.04°)              |                     |
| ERA5             | ECMWF reanalyses (NWP-based estimates)             | 1 h                 | Roughly 18 km x 28  | 5 days              |
|                  |                                                    |                     | km* (0.25° x 0.25°) |                     |
| WRF              | WRF reanalyses (with initial conditions from       | 1 h                 | 1.0 km (settable)   | 4.5 h               |
|                  | ICON model)                                        |                     |                     |                     |
| * In the area of | 1 1 1 1                                            | I                   | 1                   | L                   |

<sup>\*</sup> In the area of the study basin.

244245

# 3.2. Operational data available in real time

All measurement data require quality control (QC) employing adequately designed systems, which are often very sophisticated (Szturc et al., 2022), especially for weather radar data. These systems are dedicated to verifying the data and, if necessary, correcting them. Using different precipitation information and a cross-check approach in a QC scheme is a common practice.

#### 3.2.1. Rain gauge measurements

The network of telemetric rain gauges of IMGW – the NMHS in Poland – consists of about 650 stations, mainly of the tipping bucket type. There are 158 stations in the area analysed in this work (Fig. 3), which gives an average of one rain gauge per approximately 280 km<sup>2</sup>. This network is much denser in the mountains, including the Sudety Mountains than in other parts of the country, with one station per approximately 420 km<sup>2</sup>.

Precipitation measurements are transmitted in the form of 10-minute accumulations. Additionally, analogous data from the Czech Republic (CHMU – the Czech NMHS) from gauges near the Polish border are also operationally available. All data are subject to quality control by the RainGaugeQC system developed at IMGW (Ośródka et al., 2022; 2025). The point measurements are interpolated using the Ordinary Kriging method to obtain a precipitation field with 1-km resolution.

Figure 3: Locations of measurement stations in the upper and middle Odra River basin: telemetric rain gauges (blue dots), weather radars (brown triangles) with 150-km range (brown circles), commercial microwave links (black lines), and four manual rain gauges selected for more detailed analysis (larger blue dots).

#### 3.2.2. Weather radar measurements

POLRAD, IMGW's weather radar network, consists of 10 C-band, Doppler and polarimetric radars manufactured by Leonardo Germany. The network is supplemented by data from 10 radars from neighbouring countries, whose observations partially cover the territory of Poland (Fig. 3). The radar data are quality controlled with the RADVOL-QC system designed at IMGW (Ośródka et al., 2014; Ośródka and Szturc, 2022). The precipitation composite maps are generated based on the PseudoSRI products from individual radars with a merging algorithm that considers a combination of data quality

information and distance from the radar site (this was also developed at IMGW, Jurczyk et al., 2020a). The spatial resolution of the final field is 1 km x 1 km, and the temporal resolution is 10 min.

However, it should be noted that radar estimates of precipitation in mountainous areas are usually less reliable due to disturbances arising from the interaction of the radar beam with the terrain. Therefore, algorithms for the adjustment of radar-based precipitation with rain gauges are becoming more important. A mean-field bias correction is carried out individually for each radar based on a 10-min accumulation. Then, the spatial adjustment is performed based on a comparison of past radar estimates with corresponding rain gauge data to handle non-uniform bias within the radar composite domain (Jurczyk, 2020b).

The flooding area is within the range of five Polish radars, three located in the upper and middle Odra river basin – Pastewnik (PL\_PAS), Góra św. Anny (PL\_GSA) and Ramża (PL\_RAM), and in its vicinity – Poznan (PL\_POZ) and Brzuchania (PL\_BRZ). Moreover, two German radars, Protzel (GE\_PRO) and Dresden (GE\_DRE), and one Czech radar, Skalky (CZ\_SKA), partially cover the basin area.

#### 3.2.3. Satellite measurements and estimations

Satellite precipitation fields for Europe are based primarily on data from geostationary meteorological satellites of the Meteosat family, which are positioned over the equator at various longitudes. They are an important source of operational data due to their very high temporal resolution of 5 minutes and quick access of a few minutes. Their spatial resolution, which for the area of southern Poland is approx. 3.5 km x 6.0 km, is also relatively high in terms of satellite data.

Depending on the availability of additional data, it is possible to generate different satellite-based estimates in real time or in near real time, such as precipitation fields based on products generated by software developed by EUMETSAT programmes. IMGW operationally uses products generated by the software of the EUMETSAT NWC SAF (2021) programme from the visible (daytime CRR-Ph and PC-Ph products) and infrared (24-hour CRR and PC) data. On this basis, 10-min precipitation accumulation fields are estimated by IMGW software (Jurczyk et al., 2020b). These data are corrected by mean field bias with radar precipitation adjusted to rain gauge measurements. The H61B precipitation product of the EUMETSAT H SAF (2020) programme is also available, which, unlike the SAT product, is based only on data from the IR channel available 24 hours a day but is supplemented with observations from passive microwave sensors located on various meteorological satellites in low polar orbits.

#### 3.2.4. Other estimates

Measurements of signal attenuation on commercial microwave links (CMLs) allow the calculation of the integrated precipitation along a given link with a length of several to tens of kilometres (Olsson et al., 2025). The precipitation is spatially distributed along the link in proportion to the distribution of

weather radar (RAD) precipitation along this distance (Pasierb et al., 2024). There are 400 such links in the area analysed in this study (Fig. 3), which gives an average of one link per around 100 km<sup>2</sup>.

The CML-based 15-minute precipitation accumulations are spatially interpolated using inverse distance methods to obtain high-resolution 1 km x 1 km precipitation fields. The data are currently being tested at IMGW for their applicability to real-time operational applications.

#### 3.2.5. Multi-source estimates

The RainGRS model combining rain gauge, radar and satellite precipitation data is used operationally at IMGW (Jurczyk et al., 2020b), applying a conditional merging technique that is a development of the Sinclair and Pegram (2005) algorithm. This method is enhanced by involving detailed quality information assigned to individual input data. The combination algorithm is divided into two stages. At first, rain gauge data are merged with radar and satellite estimates separately, taking into account their quality. Finally, the resulting two precipitation fields are combined using weights depending on the distance from the nearest radar site and the quality of the satellite precipitation. As a result, a multi-source gauge-radar-satellite field (GRS) is received, with a spatial resolution of 1 km x 1 km and a temporal resolution of 10 min.

#### 3.3. Estimates not available in real time

#### 3.3.1. Manual rain gauge measurements

Figure 4: Locations of manual rain gauges (blue circles) and four ones selected for more detailed analysis (larger blue dots) in the upper and middle Odra River basin.

The IMGW network of manual rain gauges consists of about 641 stations. Their operation involves employing a graduated cylinder from which the observer reads the height of the rainwater column. In Poland, such gauges are used in the Hellmann standard, however, their measurements have some limitations: (i) they are point wise, (ii) they have relatively long precipitation accumulation times of, most often, 24 hours, (iii) they require measurement processing (including quality control), so they are not available in real time. The data from manual rain gauges are the closest to reality at their locations, and therefore were selected as the point reference for the 2024 flood. There are 112 such stations in the area analysed in this study (Fig. 4), one rain gauge per approximately 395 km<sup>2</sup>.

# 3.3.2. Satellite-based reanalyses

Satellite-based reanalyses use additional information, especially from satellites on polar low Earth orbits, beyond what is available from geostationary satellites, and this improves their reliability. However, this requires more time to acquire and process data, so the delay in access to the estimates in such cases can be as long as several months (Berthomier and Perier, 2023).

The Integrated Multi-satellitE Retrievals for GPM (IMERG) is a NASA product estimating global surface precipitation rates at a spatial and temporal resolution of 0.1° x 0.1° and 30 min, respectively (NASA, 2025). This product is calibrated with Global Precipitation Measurement (GPM Core Observatory) satellite data, which is based on microwave imager and the dual-frequency precipitation radar, and uses it as a baseline. It is combined with other observations from national or international satellite constellations equipped with weather radars and passive microwave and infrared sensors, as well as with rain gauge data (Huffman et al., 2020; Bogerd et al., 2021). IMERG has three runs with different delays: Early (4-hour delay), Late (14-hour) and Final (about 4 months).

The PERSIANN Dynamic Infrared Rainfall Rate Near Real-Time (PDIR-Now) is a global, high-resolution (0.04° x 0.04°) satellite-based precipitation estimation product developed by the University of California, Irvine (UCI) (Nguyen et al., 2020a; 2020b; Afzali Gorooh et al., 2022) (CHRS, 2025). It is based on high-frequency sampling of infrared imagery and has a timeliness of 30-60 minutes. PDIR-Now considers errors due to the use of IR imagery by applying various techniques, including dynamic curve shifting (Tb-R) based on precipitation climatology. Its highest temporal resolution is 1 hour.

## 3.3.3. Reanalyses of the NWP models

The ERA5 fields (ECMWF Reanalysis v5) generated by the ECMWF (European Centre for Medium-Range Weather Forecasts) have a low resolution of 0.25° x 0.25°, which converted to distance units corresponds to grids of approximately 18 km x 26 km in Poland (ECMWF, 2025). Such data allows for an overall analysis of rainfall offline. However, it is impossible to use these reanalyses when knowledge of the course of convective phenomena at the microscale is needed, i.e. with a spatial resolution of 1 km or less.

The WRF (Weather Research and Forecasting) is a model developed at NCAR (National Center for Atmospheric Research) (NCAR, 2025). Initial conditions for simulations of precipitation during the flood analysed here were taken from the ICON-EU (Icosahedral Nonhydrostatic) model (6.5 km) developed at Deutscher Wetterdienst (German NMS, <u>DWD, 2025</u>). Simulations were conducted at 50 vertical levels up to 50 hPa, with a horizontal resolution of 1 km and a time step of 1 hour. Thompson's microphysics scheme (Thompson et al., 2004) was utilised in the simulations. Due to the high resolution of the computational domain, explicit wet process physics was implemented, along with the parameterisation of short-wave and long-wave radiation based on the RRTMG radiation propagation scheme, a newer version of RRTM (Iacono et al., 2008). Boundary layer processes were modelled according to the Mellor-Yamada-Nakanishi-Niino (MYNN) turbulence scheme with closure 2.5 (Nakanishi and Niino, 2009). The near-surface layer was parameterised using the MYNN scheme (Nakanishi and Niino, 2006). The multi-physics Noah land surface model (Niu et al., 2011) predicts soil moisture and temperature at four depths (Jarvis, 1976).

#### 4. Reliability analysis of different techniques of precipitation measurement and estimation

## 4.1. Methodology for verifying precipitation data

The basic analyses were carried out for 1-day accumulations with reference data from manual rain gauges (GAU Manual), which we consider to be the most reliable values. These measurements are point wise, so verification of individual precipitation fields was performed only at the locations of these stations (112 ones). The data were from 13-16 September 2024, but at IMGW, measurements of meteorological daily precipitation are made at 6 UTC, i.e. the accumulation for a given day is summed from 6:00 UTC of the previous day to 6:00 UTC of the following day and assigned to the date on which the accumulation ended. Thus, the period analysed included precipitation from 6 UTC 12 September to 6 UTC 16 September.

The temporal distribution of heavy precipitation plays a key role, so the data available with a 1-hour time step was also verified. As measurements from manual rain gauges are not available at such a short time step, the RainGRS (GRS) fields (44,218 pixels within the basin) were used as a benchmark for the verification. In this case, it was possible to conduct a spatial verification because the reference was data with a resolution of 1 km x 1 km. However, it should be noted that the GRS estimates depend on some of the verified data (GAU, RAD, RAD Adj, and SAT).

The following metrics were employed:

 Pearson correlation coefficient is a well-known metric which is sensitive to a linear relationship between two datasets and reflects agreement between estimate and reference in terms of spatial pattern:

$$CC = \frac{\sum_{i=1}^{n} (E_i - \overline{E})(o_i - \overline{o})}{\sqrt{\sum_{i=1}^{n} (o_i - \overline{o})^2 \sum_{i=1}^{n} (E_i - \overline{E})^2}}$$
 (1)

 root mean square error based on variance is a standard metric used in verification studies as a good measure of differences between the verified and reference values:

$$RMSE = \sqrt{\frac{1}{n} \sum_{i=1}^{n} (E_i - O_i)^2}$$
 (2)

The RMSE is particularly sensitive to outliers as squaring the errors emphasizes larger deviations.

 root relative square error is similar to RMSE, but it is scale-independent as it relates the deviations to the spread of the reference values around their mean:

$$RRSE = \frac{\sqrt{\sum_{i=1}^{n} (E_i - O_i)^2}}{\sqrt{\sum_{i=1}^{n} (O_i - \overline{O})^2}}$$
 (3)

statistical bias, which is a measure of systematic error:

Bias =
$$\frac{1}{n} \sum_{i=1}^{n} (E_i - O_i)$$
 (4)

where  $E_i$  is the estimated value,  $O_i$  is the reference value, i is the gauge/pixel number, and n is the number of gauges/pixels, whereas  $\overline{E}$  and  $\overline{O}$  are the mean values of  $E_i$  and  $O_i$ , respectively.

# 4.2. Precipitation fields obtained from various measurement techniques and estimation methods

- Daily precipitation accumulations for the flood event of 13-16 September 2024, derived from various measurement techniques and estimation methods described in this paper (Table 1), are presented below: (i) reference data from spatially interpolated manual rain gauge observations (Fig. 5), (ii) precipitation fields operationally available in real time (Fig. 6), and (iii) offline reanalyses (Fig. 7).
- A visual assessment of the differences between all the verified data and the reference allows the following general observations to be formulated.
- The GAU and multi-source GRS rain gauge fields accurately reproduce the spatial distribution of the precipitation field and are consistent with the reference in terms of values. Differences are visible mainly in the Karkonosze Mountains on the border with the Czech Republic, probably due to the

densities of the GAU Manual and GAU networks (the latter is higher in this area) and the influence of data from the Czech territory.

In the case of radar-derived fields (RAD and RAD Adj), the precipitation pattern is also well represented, but the estimate based solely on radar observations (RAD) underestimates values. Therefore, unadjusted radar data should not be used, especially for quantitative precipitation estimates (WMO-No. 1257, 2025). Radar data after adjustment with rain gauge measurements (RAD Adj) demonstrates good agreement concerning precipitation values.". The radar network in the analysed flood area is relatively dense, but due to signal blocking by mountains, precipitation shadows appear in some places, which result in an underestimation of precipitation. This is particularly evident in the Kłodzko Valley which is surrounded by relatively high mountains and is one of the places most prone to catastrophic flooding.

Estimates generated based on satellite data: SAT, H61B and PDIR-Now, reproduce the precipitation distribution in space very imprecisely and values are significantly lower than the reference. The IMERG reanalysis definitely represents the precipitation field better, but values are also underestimated, especially in places where accumulations are highest. The reliability of precipitation estimates based on satellite data is low, especially when they are generated from infrared channel data and are not supported by other, preferably microwave data (from radars). This mainly affects SAT estimates, but also others. It should be noted that during the analysed flood, data from visible channels was only available for about 1/3 of the time, due to the fact that for the measurements to be reliable, the sun must be sufficiently high above the horizon (above 20 degrees). Furthermore, the spatial resolution of these data is generally insufficient.

The CML-based estimates represent precipitation variability quite correctly, but the values compared to the reference are slightly lower. It can be clearly seen that spatial representativity is limited due to the lower density of the links in the higher parts of the mountains, such as in the eastern part of the Kłodzko Valley.

Estimates based on numerical mesoscale models (ERA5 and WRF) correctly reproduce the precipitation pattern. However, the ERA5 reanalyses have a very low spatial resolution, so they do not reflect the fine-scale structures of the precipitation field, and, in addition, the values are more underestimated than those derived from WRF simulations.

Figure 6: Precipitation fields available in real time for daily precipitation accumulations from 13-16 September 2024. Data are limited to the upper and middle Odra river basin area.

Figure 7: Precipitation fields available offline for daily precipitation accumulations from 13-16 September 2024. Data are limited to the upper and middle Odra river basin area.

# 4.3. Verification of daily and hourly precipitation accumulations

Daily precipitation accumulations derived from different measurement techniques and estimations, listed in Table 1, were verified against point-wise observations from manual rain gauges. Table 2 summarises the values of the characteristics defined in Section 4.1 and, additionally, the relationship between CC and RMSE values for the verified measurement techniques is shown in the graph in Fig. 8.

Table 2. Values of statistics for daily precipitation accumulations from 13-16 September 2024, against data from manual rain gauges (GAU Manual) as reference.

| Measurement/estimation | Mean      | CC()         | RMSE  | RRSE | Bias   |  |  |  |  |  |  |
|------------------------|-----------|--------------|-------|------|--------|--|--|--|--|--|--|
| technique              | (mm)      | CC (-)       | (mm)  | (-)  | (mm)   |  |  |  |  |  |  |
| Reference data         |           |              |       |      |        |  |  |  |  |  |  |
| GAU Manual             | 41.78     | -            | -     | -    | -      |  |  |  |  |  |  |
|                        | Available | in real time |       |      |        |  |  |  |  |  |  |
| GAU                    | 38.27     | 0.963        | 10.40 | 0.29 | -3.50  |  |  |  |  |  |  |
| RAD                    | 16.07     | 0.784        | 38.08 | 1.06 | -25.71 |  |  |  |  |  |  |
| RAD Adj                | 36.65     | 0.956        | 12.42 | 0.35 | -5.13  |  |  |  |  |  |  |
| SAT                    | 10.02     | 0.395        | 46.06 | 1.28 | -31.76 |  |  |  |  |  |  |
| H61B                   | 18.77     | 0.455        | 39.46 | 1.10 | -23.00 |  |  |  |  |  |  |
| CML                    | 21.13     | 0.721        | 32.74 | 0.91 | -20.65 |  |  |  |  |  |  |
| GRS                    | 37.94     | 0.967        | 10.02 | 0.28 | -3.83  |  |  |  |  |  |  |
|                        | Availa    | ble offline  | l     |      |        |  |  |  |  |  |  |
| IMERG                  | 27.15     | 0.552        | 33.40 | 0.93 | -14.63 |  |  |  |  |  |  |

| PDIR-Now | 20.23 | 0.138 | 42.57 | 1.18 | -21.55 |
|----------|-------|-------|-------|------|--------|
| ERA5     | 32.63 | 0.748 | 26.00 | 0.72 | -9.15  |
| WRF      | 30.48 | 0.759 | 26.02 | 0.72 | -11.30 |

Most of the analysed data estimate precipitation correctly, in particular the GAU, RAD Adj, and GRS fields, which exhibit an extremely high correlation coefficient (CC > 0.9), and the differences between verified and reference values are very low taking into account the magnitude of the rainfall (RMSE < 15 mm). Therefore, these fields can correctly represent precipitation with high spatial resolution for operational purposes and subsequent analyses.

quite good agreement with the reference, but RMSE is already high, above 25 mm. WRF reanalyses

turned out better with CC = 0.77 and RMSE = 25.6 mm. In the case of the RAD and CML fields, the correlation coefficient is also high (CC > 0.7), but a significant underestimation of precipitation is

which CC < 0.5 and RMSE > 35 mm, and only slightly better statistics were achieved for the IMERG

evident, as indicated by large RMSE values > 30 mm, with Bias of -25.7 and -20.6, respectively.

The ERA5 and WRF simulations performed slightly worse, with CC above 0.7, which suggests

The worst results were obtained for the satellite-based estimates: SAT, H61B and PDIR-Now, for

estimates (CC = 0.55, RMSE = 33.4 mm).

490

492

> 1.0 0.9 WRF 8.0 8 RAD 0.7 ERA5 CML 0.6 O 0.5 **IMERG** H61B 0.4 SAT 0.3 0.2 0.1 PDIR-Now 0.0 0 5 10 15 20 25 30 35 40 45 50

494

Figure 8: Scatter plot comparing CC vs RMSE for each measurement and estimation technique for daily precipitation accumulations from 13-16 September 2024, against data from manual rain gauges (GAU Manual) as reference.

RMSE (mm)

497

499 Further research was conducted to evaluate the usefulness of the investigated data at a higher temporal resolution – hourly instead of daily. Table 3 shows results analogous to those depicted in Table 2, but the reference in this case are the RainGRS estimates (GRS fields), as measurements from manual

rain gauges are only available as daily accumulations. This data was selected as a benchmark because the correlation between the two fields (i.e. GAU Manual and GRS) for daily accumulations is the best, being as high as 0.97 and Bias is as low as -3.8 mm (Table 2). The relationship between CC and RMSE values for the verified measurement techniques is shown in the graph in Fig. 9.

Table 3. Values of statistics for hourly precipitation accumulations from 13-16 September 2024, against the RainGRS estimates (GRS) as reference.

| Measurement/estimation | Mean      | CC()         | RMSE | RRSE | Bias  |  |  |  |  |  |
|------------------------|-----------|--------------|------|------|-------|--|--|--|--|--|
| technique              | (mm)      | CC (-)       | (mm) | (-)  | (mm)  |  |  |  |  |  |
| Reference data         |           |              |      |      |       |  |  |  |  |  |
| GRS                    | 1.05      | -            | -    | -    | -     |  |  |  |  |  |
|                        | Available | in real time | !    |      |       |  |  |  |  |  |
| GAU (dependent)        | 1.03      | 0.906        | 0.60 | 0.41 | -0.01 |  |  |  |  |  |
| RAD (dependent)        | 0.49      | 0.902        | 1.07 | 0.70 | -0.56 |  |  |  |  |  |
| RAD Adj (dependent)    | 1.03      | 0.977        | 0.29 | 0.22 | -0.01 |  |  |  |  |  |
| SAT (dependent)        | 0.33      | 0.256        | 1.75 | 1.21 | -0.72 |  |  |  |  |  |
| H61B                   | 0.61      | 0.174        | 1.70 | 1.21 | -0.44 |  |  |  |  |  |
| CML                    | 0.60      | 0.673        | 1.11 | 0.83 | -0.45 |  |  |  |  |  |
|                        | Availa    | ble offline  |      |      |       |  |  |  |  |  |
| IMERG                  | 0.94      | 0.529        | 1.39 | 0.98 | -0.11 |  |  |  |  |  |
| PDIR-Now               | 0.70      | 0.114        | 1.89 | 1.45 | -0.35 |  |  |  |  |  |
| ERA5                   | 1.11      | 0.497        | 1.34 | 0.93 | 0.06  |  |  |  |  |  |
| WRF                    | 0.94      | 0.367        | 1.67 | 1.20 | -0.10 |  |  |  |  |  |

In terms of the much higher temporal resolution of the measurements and estimates, fewer of them maintain a correspondingly high reliability. Both the GAU and RAD Adj estimates demonstrated excellent results, with CC values exceeding 0.9 and RMSE values of 0.6 mm and 0.3 mm, respectively. The raw radar data (RAD) also correlates well with the reference, achieving CC of 0.90; however, the discrepancies between values are larger, resulting in RMSE of 1.1 mm. It is important to note that the GRS products depend on all three data fields.

Among the other data not involved in multi-source RainGRS combination, relatively high reliability was preserved by the CML field with the best correlation coefficient (CC = 0.67), but Bias is significant (Bias = -0.4) even though RMSE is not relatively high (RMSE = 1.1 mm). Model simulations ERA5 and WRF do not correlate well with the reference (CC = 0.50 and 0.37, respectively), and the discrepancy in value is large (RMSE are 1.3 and 1.7 mm, respectively).

IMERG analyses proved to be the most reliable satellite-based products compared in this work. By incorporating multiple precipitation data sources, which takes several months, a correlation with reference (CC = 0.53) is better than both model simulations but worse than that obtained by rain gauge,

radar measurements, and even CMLs. The statistics for the other satellite-based estimates (SAT, H61B, and PDIR-Now) turned out to be much worse: CC < 0.26 and RMSE > 1.7 mm, moreover, they drastically underestimate rainfall (their negative Bias is more than 0.35 mm).

Figure 9: Scatter plot comparing CC vs RMSE for each measurement and estimation technique for hourly precipitation accumulations from 13-16 September 2024, against the RainGRS estimates (GRS) as reference.

# 4.4. Verification of extreme daily and hourly precipitation accumulations

For effective flood protection, it is important to have accurate values of very high precipitation. In order to assess the reliability of the measurements and estimations of extreme accumulations, verification was conducted by introducing a threshold on the minimum reference precipitation value.

The results of the statistical analysis based on daily accumulations from manual rain gauge measurements (GAU Manual) for days with recorded rainfall of 50 mm or more are presented in Table 4. The relationship between CC and RMSE values for the verified measurement techniques is shown in the graph in Fig. 10.

As expected, the results are noticeably worse when compared to those obtained without a limitation on precipitation magnitude (see Table 2). This is particularly evident in terms of bias, which indicates an increase in underestimation. However, a negative bias was observed for all the estimation techniques analysed, even without thresholding. This suggests a real underestimation of intense precipitation by these methods, rather than simply a result of data selection. Excellent agreement with the reference high precipitation was obtained by rain gauge observations (GAU) and estimates directly based on measurements (RAD Adj and GRS) for which CC > 0.85 and RMSE < 25 mm. The estimate

based solely on radar data (RAD) correlates quite well (CC = 0.61), but the values are strongly underestimated (RMSE = 63.8 mm, Bias = -58.1 mm).

Table 4. Values of statistics for daily precipitation accumulations from 13-16 September 2024 against data from manual rain gauges (GAU Manual) as a reference with a threshold for daily precipitation of 50 mm.

| Measurement/estimation | Mean      | CC()         | RMSE  | RRSE | Bias   |  |  |  |  |  |
|------------------------|-----------|--------------|-------|------|--------|--|--|--|--|--|
| technique              | (mm)      | CC (-)       | (mm)  | (-)  | (mm)   |  |  |  |  |  |
| Reference data         |           |              |       |      |        |  |  |  |  |  |
| GAU Manual             | 84.38     | -            | -     | -    | -      |  |  |  |  |  |
|                        | Available | in real time | ,     |      |        |  |  |  |  |  |
| GAU                    | 76.90     | 0.889        | 16.66 | 0.53 | -7.49  |  |  |  |  |  |
| RAD                    | 26.30     | 0.614        | 63.76 | 2.03 | -58.08 |  |  |  |  |  |
| RAD Adj                | 70.81     | 0.880        | 20.18 | 0.64 | -13.57 |  |  |  |  |  |
| SAT                    | 14.43     | 0.413        | 75.63 | 2.40 | -69.95 |  |  |  |  |  |
| H61B                   | 28.42     | 0.283        | 64.20 | 2.04 | -55.96 |  |  |  |  |  |
| CML                    | 42.06     | 0.301        | 53.58 | 1.70 | -42.32 |  |  |  |  |  |
| GRS                    | 75.89     | 0.904        | 16.09 | 0.51 | -8.49  |  |  |  |  |  |
|                        | Availa    | ble offline  |       |      |        |  |  |  |  |  |
| IMERG                  | 39.75     | 0.336        | 54.82 | 1.74 | -44.64 |  |  |  |  |  |
| PDIR-Now               | 23.49     | 0.170        | 68.81 | 2.19 | -60.89 |  |  |  |  |  |
| ERA5                   | 54.33     | 0.357        | 43.28 | 1.37 | -30.05 |  |  |  |  |  |
| WRF                    | 56.51     | 0.479        | 41.14 | 1.31 | -27.87 |  |  |  |  |  |

All satellite-based data are inconsistent with the benchmark, as indicated by the low correlation (CC < 0.42) and significant differences in precipitation values (RMSE > 50 mm). The IMERG product also has low reliability, although it outperformed the other satellite-derived estimates in previous verifications.

The result of the verification of the CML estimates is quite surprising compared to the earlier ones: they have a relatively low correlation (CC = 0.30) and a rather high RMSE (53.6 mm). This can be explained by the non-uniform distribution of transmitting and receiving stations: in the mountains – where the highest precipitation was recorded – their network is much sparser compared to other areas (the opposite in the case of rain gauge networks).

The ERA5 and WRF model simulations have similar errors on precipitation values (RMSE  $\sim$  42 mm), but the correlation is a bit better for the WRF model (CC = 0.48), which may be due to the much higher spatial resolution of this model. In previous verifications (Table 2), models achieved comparable results regarding both CC and RMSE. The models still outperform satellite-based estimates.

Figure 10: Scatter plot comparing CC vs RMSE for each measurement and estimation technique for daily precipitation accumulations from 13-16 September 2024 against data from manual rain gauges (GAU Manual) as a reference with a threshold of 50 mm.

A similar analysis was conducted, but the reliability of measurements and precipitation estimates for high precipitation were verified using hourly accumulations instead of daily accumulations. The results are depicted in Table 5 and in the graph in Fig. 12. In this case, the reference dataset consists of RainGRS (GRS) estimates, applying a threshold for hourly precipitation accumulation of 5 mm, with the assumption that there must be at least 200 pixels (out of a total of 44,218 pixels) fulfilling this requirement in a given time step. Thresholds of 5-mm for hourly accumulations and 200 pixels for the area where such precipitation occurred (approximately 0.5% of the entire basin) were introduced to exclude data with low precipitation from the statistics. Fig. 11 shows, as an example, the multi-source GRS hyetogram at the Kamienica manual rain gauge location, which recorded the highest 4-day precipitation accumulation of all stations in the flood area.

Figure 11: Hyetogram of 1-hour RainGRS (GRS) estimates at the location of the Kamienica rain gauge station. The red line indicates the 5-mm threshold of hourly precipitation accumulations.

In this verification, the statistical results are significantly worse than in Table 3, as correctly reproducing extremely high hourly precipitation accumulations is challenging. Only GAU, RAD, and RAD Adj measurements provide relatively reliable results regarding correlation with GRS (CC > 0.50). As in the previous analyses, the estimate based solely on RAD data gives a significant underestimation of rainfall (RMSE = 4.3 mm, Bias = -4.1 mm), while for the fields based on rain gauge data, these errors are much lower: RMSE for GAU and RAD Adj is 2.5 and 0.8 mm, respectively. However, it is important to note that the GRS reference depends on all estimates using rain gauge or radar data.

Table 5. Values of statistics for hourly precipitation accumulations from 13-16 September 2024 against the RainGRS estimates (GRS) as a reference with a threshold for hourly precipitation of 5 mm.

| Measurement/estimation | Mean      | CC()         | RMSE | RRSE | Bias  |  |  |  |  |  |
|------------------------|-----------|--------------|------|------|-------|--|--|--|--|--|
| technique              | (mm)      | CC (-)       | (mm) | (-)  | (mm)  |  |  |  |  |  |
| Reference data         |           |              |      |      |       |  |  |  |  |  |
| GRS                    | 7.03      | -            | -    | -    | -     |  |  |  |  |  |
|                        | Available | in real time | -    |      |       |  |  |  |  |  |
| GAU (dependent)        | 5.29      | 0.515        | 2.46 | 1.63 | -1.75 |  |  |  |  |  |
| RAD (dependent)        | 2.96      | 0.630        | 4.32 | 2.92 | -4.08 |  |  |  |  |  |
| RAD Adj (dependent)    | 7.02      | 0.907        | 0.76 | 0.57 | -0.01 |  |  |  |  |  |
| SAT (dependent)        | 0.85      | 0.089        | 6.60 | 4.68 | -6.19 |  |  |  |  |  |
| H61B                   | 1.21      | 0.029        | 6.33 | 4.41 | -5.83 |  |  |  |  |  |
| CML                    | 3.37      | 0.269        | 4.28 | 2.96 | -3.66 |  |  |  |  |  |
| Available offline      |           |              |      |      |       |  |  |  |  |  |
| IMERG                  | 2.60      | 0.069        | 5.16 | 3.42 | -4.44 |  |  |  |  |  |
| PDIR-Now               | 1.06      | 0.046        | 6.40 | 4.43 | -5.97 |  |  |  |  |  |
| ERA5                   | 2.27      | 0.062        | 5.24 | 3.57 | -4.77 |  |  |  |  |  |

| WRF | 2.38 | 0.069 | 5.54 | 3.84 | -4.66 |
|-----|------|-------|------|------|-------|
|     |      |       |      |      |       |

Among the datasets not involved in multi-source RainGRS estimation, none of the correlations exceed CC = 0.1 except for the CML estimate (CC = 0.27). The values of RMSE and Bias are also high for them (RMSE > 5 mm, Bias between -4 and -6).

The conclusion from this analysis is that the estimation of extremely high precipitation fields with very high spatial (1 km) and temporal (1 hour) resolution is mainly based on weather radar observations, but these must first be adjusted to the rain gauge data. Rain gauges can also produce reliable estimates, but under the condition that a sufficiently dense network of such gauges is available.

Figure 12: Scatter plot comparing CC vs RMSE for each measurement and estimation technique for hourly precipitation accumulations from 13-16 September 2024 against the RainGRS estimates (GRS) as a reference with a threshold of 5 mm.

# 4.5. Analyses for selected stations

 Four stations with manual rain gauges (GAU Manual) were selected to check the consistency of the precipitation estimated by different techniques and models concerning particular locations for four days with the highest values during the flood. They are located in different regions of the basin, where intense rainfall was observed (Fig. 4), moving from west to east of the Sudety Mountains:

- Szklarska Poręba in the Karkonosze Mountains,
- Kamienica in the Śnieżnik Mountains near the Kłodzko Valley (the highest daily as well as 4day precipitation was observed there during this flood),
- Głuchołazy situated in the foothills of the Opawskie Mountains,

Gołkowice located in the Ostrava Valley.

Table A1 (in Appendix A) presents accumulations for all verified measurements and estimates for individual days and the four-day totals in these locations. In Szklarska Poręba and Kamienica, telemetric rain gauges (GAU) measured daily accumulations very close to the reference rainfall (GAU Manual), while the other two locations underestimated by about 10-20%. The daily distribution of RAD values indicates good temporal alignment with the GAU Manual, but a significant underestimation of rainfall is evident. Adjustment of the radar-based estimates to rain gauge measurements resulted in a significant increase in RAD Adj values, but they are still lower than the GAU Manual at all locations except Gołkowice. GRS precipitation accumulations for three stations (Szklarska Poręba, Kamienica and Głuchołazy) are similar to GAU, i.e. also underestimated in relation to the reference by about 10-20%. At the Gołkowice location, where there is no telemetric rain gauge, and the GAU values are derived from interpolation, the GRS estimates are very close to the RAD Adj values and overestimate the benchmark.

Estimates based on CML data are significantly lower than the reference, except Szklarska Poręba, where the density of the microwave link network is relatively high. This underestimation in the other stations is probably due to the lack of links near them, so values are derived from the interpolation of slightly more distant links, usually located at lower altitudes, which record less precipitation.

The variability of all satellite-based precipitation in the analysed days does not correspond well with the daily distribution of the reference. Accumulations are much lower in comparison to values measured by manual rain gauges. The IMERG reanalyses slightly outperform the others, which is similar to previous investigations.

Mesoscale model simulations are also underestimated, although the WRF model does so to a lesser extent. They better reflect the temporal distribution of daily precipitation accumulations and their magnitudes than satellite data.

The cumulative precipitation curves obtained from 1-hour accumulations for the same four stations are shown in Fig. 13. The GAU Manual data generated with a daily step were not included, and in consequence, the GRS estimates (see Section 4.3) were taken as a reference to assess the consistency of temporal distributions of verified precipitation. It can be seen from analyses of the curves for all four stations that the estimates on which the GRS data depend, i.e. those based on rain gauge and radar measurements, are similar to each other, although the differences between the reference and the values derived solely from radars observations are very large. In terms of the independent data, the curves for CML and WRF reflect the temporal distribution of precipitation relatively correctly. In contrast, all satellite-based estimates are highly inconsistent with the reference, taking into account precipitation variability in time, and among them, the IMERG reanalyses indicate the best temporal alignment, as in previous investigations.

Figure 13: Cumulative hourly precipitation accumulations for the four stations from Table 6 for the period 13-16 September 2024.

# 4.6. Overall assessment of the various rainfall measurement techniques

The evaluation of the results obtained in this study is mainly based on the numerical values summarised in Tables 2 to 5, where the reliability statistics of the individual measurements and estimations are shown. The analyses were conducted with daily accumulations from the GAU Manual (Tables 2 and 4) and 1-hour RainGRS estimates as references (Tables 3 and 5). It should be noted that the latter depends, to differing degrees, on data involved in multi-source combination GAU, RAD, and RAD Adj, and to a lesser extent on SAT product. Nevertheless, the proportions between the statistics' values are similar using both references. This leads to the conclusion that this dependence has little influence on the final outcomes, however the following overall assessment does not include findings from the analysis of the consistency of individual data with the reference dependent on them.

#### 4.6.1. Rain gauge data

Spatially interpolated telemetric precipitation data (GAU) proved to be very similar to measurements from manual rain gauges (GAU Manual), but they generally provide slightly lower values (Tables 2 and 4). The accuracy of the rain gauge observations also remains high if only heavy precipitation is considered, which is confirmed by the statistics calculated after introducing an appropriate threshold on the daily accumulations, as can be seen from a comparison of Table 4 and Table 2.

Notably, 76 out of 158 telemetric rain gauges are in the same locations as manual ones in the flood area. This significantly impacts the reliability statistics calculated for the GAU data as, in the case of an interpolated field, estimated values strongly depend on the distance to the nearest station.

#### 4.6.2. Weather radar-based data

Weather radars reflect the spatial and temporal distributions of the precipitation field very well, as evidenced by the very high CC correlation coefficients with the reference presented in all tables, especially Tables 2 and 4, where the benchmark data are independent of the radar measurements.

Raw radar estimates RAD produced significantly underestimated precipitation values, as indicated e.g. by the very large Bias values (Tables 2 and 4). Adjusting with telemetric rain gauge data considerably improves this and makes the corrected radar-based precipitation field (RAD Adj) very close in precipitation values to both GAU Manual and GRS reference estimates.

Analysing only high precipitation, i.e. after introducing an appropriate threshold on the amount of daily precipitation accumulation, the results were analogous to the analysis without applying a threshold (Tables 4 vs 2). This confirms the high reliability of the radar measurements also in the case of heavy precipitation, however the data without adjustment is subject to a large Bias.

#### 4.6.3. Satellite-based data

The satellite-based real-time SAT and H61B fields, based on the products from the EUMETSAT NWC SAF and H SAF programmes respectively, turned out to be practically useless for the precipitation estimation in the case study analysed here. They correlate poorly with reference and significantly underestimate values of precipitation accumulation (Tables 2 and 3). The primary reason is that they are mainly based on data from geostationary satellites – the only kind that can be used directly for real-time measurements at high temporal resolution. Among the more advanced satellite-based precipitation products available only offline analysed in this work, it can be stated that the PDIR-Now estimates are definitely wrong. The IMERG reanalysis proved significantly better, although its reliability is also not high.

If the highest daily accumulations are considered by limiting them to values above the threshold of 50 mm per day, only SAT precipitation based on NWC SAF products shows some agreement with the reference, although it is weak (Table 4). The correlations of all satellite estimates decrease dramatically for extreme 1-hour accumulations (Table 5).

#### 4.6.4. Multi-source estimates

The multi-source GRS estimates are generated by the RainGRS system for the merging GAU, RAD Adj, and SAT precipitation measurements. The analyses carried out in this study showed that these fields, among all the verified data available in real time, are in the best agreement with independent reference observations from manual rain gauges (GAU Manual) (Tables 2 and 4). The metrics are

slightly better than those for spatially interpolated rain gauges, but the multi-source estimates significantly outperform the others. This results from the combination that utilises the individual inputs' positive features (see Sects. 1.2.2 and 3.2.5).

#### 4.6.5. CML-based estimates

CML-based estimates correlate relatively well with daily and hourly accumulation benchmarks, but relatively high errors relate to differences between verified and reference values: RMSE and Bias. Data estimated from the measurements of signal attenuation from commercial microwave links in precipitation are clearly better than satellite-derived fields, even those available offline, but they are worse than estimates based on rain gauge and radar information. Their reliability is similar to mesoscale model simulations in terms of daily data, however for hourly accumulations the CML-based estimates outperform them (Tables 2 and 3). This suggests better representativeness in the temporal distribution of precipitation.

These relatively good statistics for CML-based data are probably because the network of links is very dense relative to the rain gauge network, which partly compensates for their much higher uncertainty. However, there are considerably fewer links in the highest, less urbanised mountainous areas, where precipitation is usually more intense and the detection of extreme precipitation is consequently subject to more significant errors (Tables 4 and 5).

#### 4.6.6. NWP-based reanalyses

The NWP simulations have higher reliability than satellite data but clearly lower than radar and rain gauge measurements. Their metrics are similar when analysing daily accumulations (Table 2), whereas for hourly ones, they turned out worse in comparison with CML-based data (Table 3).

The results obtained by the ERA5 and WRF models are ambiguous. In terms of daily accumulation investigations, the reliability of both models is comparable. When analysing 1-hour data (Table 3), the ERA5 reanalyses proved to be better, although their CC is not high, which indicates a more correct alignment of the precipitation variability in time. In turn, the WRF model performed better if the highest daily accumulations were considered, i.e. only above 50 mm per day (Table 4). This is probably due to the significantly (around 20 times) higher spatial resolution of the WRF model compared to ERA5, which increases their usefulness for detailed analyses of precipitation more variable in space. When it comes to extreme hourly precipitation, i.e. with a threshold for precipitation above 5 mm, none of the mesoscale models are reliable: correlations with the GRS field do not exceed CC = 0.10 for both (Table 5).

#### 5. Conclusions

In this work, detailed analyses were carried out of the reliability of different precipitation measurements and estimations during a large flood in Poland in 2024 caused by extremely high widespread precipitation in an orographically diversified basin.

Their consistency was assessed with the precipitation field or point observations assumed to be closest to reality (ground truth). As a reference, data from manual rain gauges (GAU Manual) were chosen as they are considered to be the most accurate, but they are point-wise and have the limitation of a temporal resolution of 1 day. In order to test the usefulness of data with a higher 1-hour temporal resolution, RainGRS estimates (GRS) were used as a benchmark. In addition, similar analyses were conducted, but only the most intense precipitation was considered by applying appropriate thresholds (over 50 mm/day and 5 mm/hour).

Comparing the various precipitation fields available in real time, the data based on telemetric rain gauge measurements (GAU) and weather radar observations after adjustment with rain gauge data (RAD Adj), as well as the multi-source estimates (GRS) derived from a combination of these two types of data supplemented with satellite information, are definitely most reliable. It can be concluded that during intense precipitation events triggering floods, even in mountainous areas, rain gauge and radar measurements are sufficient for accurate real-time monitoring of the precipitation field with high spatial and temporal resolution, even though IMGW's measurement networks are not very dense compared to those of other European countries.

Among the other precipitation data sources, CML-based estimates proved to be the most accurate. This is surprising as they are based on non-standard measurements, but their strength is the very high number of microwave links available. However, these data show a large underestimation of precipitation, indicating the need for more sophisticated quality control and unbiasing.

Reliability analyses of satellite data show that they are generally of little usefulness, apart from the IMERG estimates. Their relatively good agreement with the reference is due to incorporating a higher number of different types of satellite measurements, mainly microwave. However, this involves long waiting times for the final estimates which rather excludes them from operational applications, though they can be helpful in reanalyses.

The research showed the limited suitability of mesoscale model simulations for analyses with high temporal and spatial resolution. At the same time, their reliability is sufficient for use when such a requirement is not necessary. Consequently, they are not particularly useful for analyses of very intense and spatially variable precipitation.

#### Appendix A

Table A1. Comparison of daily and 4-day precipitation accumulations for 4 selected stations at locations of
manual rain gauges (Szklarska Poręba, Kamienica, Głuchołazy, Gołkowice).

|                        | S     | Station: Szklarska Poręba |         |           |        | Station: Kamienica |        |        |       |        |
|------------------------|-------|---------------------------|---------|-----------|--------|--------------------|--------|--------|-------|--------|
| Measurement/estimation | 13    | 14                        | 15      | 16        | 4-day  | 13                 | 14     | 15     | 16    | 4 day  |
| technique              | Sept  | Sept                      | Sept    | Sept      | 4-uay  | Sept               | Sept   | Sept   | Sept  | 4-day  |
| Reference data         |       |                           |         |           |        |                    |        |        |       |        |
| GAU Manual             | 20.0  | 123.8                     | 84.9    | 63.5      | 292.2  | 51.1               | 114.2  | 254.5  | 52.7  | 472.5  |
|                        |       |                           | Availab | le in rea | l time |                    |        |        |       |        |
| GAU                    | 21.72 | 131.15                    | 85.11   | 50.47     | 288.45 | 49.60              | 120.2  | 236.82 | 51.39 | 458.01 |
| RAD                    | 10.27 | 40.51                     | 22.94   | 12.71     | 86.43  | 26.09              | 27.06  | 52.08  | 12.29 | 117.52 |
| RAD Adj                | 13.57 | 120.53                    | 69.03   | 38.96     | 242.09 | 48.08              | 80.08  | 179.46 | 40.12 | 347.74 |
| SAT                    | 23.22 | 14.51                     | 7.24    | 9.30      | 54.27  | 9.19               | 41.90  | 30.59  | 5.16  | 86.84  |
| H61B                   | 24.63 | 37.53                     | 78.68   | 0.42      | 141.26 | 23.90              | 57.44  | 29.77  | 6.28  | 117.38 |
| CML                    | 29.89 | 136.66                    | 56.68   | 36.03     | 259.26 | 2.50               | 22.38  | 40.24  | 24.77 | 89.89  |
| GRS                    | 20.06 | 130.96                    | 79.72   | 47.38     | 278.12 | 50.61              | 118.06 | 227.15 | 48.71 | 444.53 |
|                        |       |                           | Avail   | able off  | line   |                    |        | II.    |       | 1.     |
| IMERG                  | 31.85 | 58.41                     | 15.22   | 14.27     | 119.75 | 40.54              | 61.81  | 39.81  | 18.72 | 160.88 |
| PDIR-Now               | 42.00 | 35.00                     | 31.00   | 4.00      | 112.00 | 29.00              | 40.00  | 19.00  | 11.00 | 99.00  |
| ERA5                   | 9.18  | 41.43                     | 12.75   | 23.17     | 86.53  | 33.51              | 67.34  | 82.21  | 24.69 | 207.75 |
| WRF                    | 1.81  | 65.93                     | 7.43    | 59.28     | 134.45 | 33.69              | 118.82 | 95.69  | 52.53 | 300.73 |

|                        | Station: Glucholazy |        |         |           | Station: Gołkowice |       |        |        |       |        |
|------------------------|---------------------|--------|---------|-----------|--------------------|-------|--------|--------|-------|--------|
| Measurement/estimation | 13                  | 14     | 15      | 16        | 4 dan              | 13    | 14     | 15     | 16    | 4 dan  |
| technique              | Sept                | Sept   | Sept    | Sept      | 4-day              | Sept  | Sept   | Sept   | Sept  | 4-day  |
|                        |                     |        | Refe    | rence da  | ata                |       |        |        |       |        |
| GAU Manual             | 56.0                | 158.2  | 124.3   | 23.0      | 361.5              | 9.7   | 96.2   | 118.2  | 3.8   | 227.9  |
|                        |                     |        | Availab | le in rea | l time             |       |        |        |       |        |
| GAU                    | 51.19               | 131.35 | 93.33   | 26.81     | 302.68             | 7.62  | 90.22  | 101.27 | 3.29  | 202.40 |
| RAD                    | 19.55               | 39.66  | 26.95   | 9.28      | 95.44              | 3.64  | 49.18  | 51.86  | 2.05  | 106.73 |
| RAD Adj                | 53.86               | 135.36 | 93.61   | 28.37     | 311.20             | 7.29  | 111.65 | 120.89 | 4.50  | 244.33 |
| SAT                    | 3.84                | 32.80  | 15.95   | 4.97      | 57.56              | 1.31  | 39.89  | 14.55  | 0.33  | 56.08  |
| H61B                   | 10.90               | 48.14  | 26.09   | 3.64      | 88.77              | 3.63  | 52.41  | 22.20  | 2.67  | 80.90  |
| CML                    | 8.95                | 34.09  | 43.32   | 11.76     | 98.12              | 2.50  | 22.14  | 63.43  | 2.11  | 90.18  |
| GRS                    | 51.82               | 133.06 | 93.40   | 26.77     | 305.05             | 7.96  | 113.61 | 118.33 | 4.26  | 244.16 |
| Available offline      |                     |        |         |           |                    |       |        |        |       |        |
| IMERG                  | 26.13               | 74.61  | 54.68   | 9.20      | 164.62             | 7.77  | 67.48  | 77.85  | 4.48  | 157.58 |
| PDIR-Now               | 15.00               | 28.00  | 21.00   | 5.00      | 69.00              | 6.00  | 39.00  | 20.00  | 6.00  | 71.00  |
| ERA5                   | 36.01               | 68.28  | 122.77  | 21.64     | 248.70             | 17.09 | 38.64  | 90.38  | 4.66  | 150.76 |
| WRF                    | 36.25               | 93.30  | 88.53   | 33.80     | 251.88             | 11.32 | 70.32  | 79.06  | 10.55 | 171.25 |

- Code availability. The data processing codes are protected by the economic property rights of the
- software and are not available for distribution. The codes used for processing follow the methodologies
- and equations described herein.

- Data availability. Out of the data used in this paper, the following are publicly available:
- Rain gauge, weather radar, RainGRS, satellite (SAT and H61B) data of IMGW:
- <a href="https://danepubliczne.imgw.pl/pl/datastore">https://danepubliczne.imgw.pl/pl/datastore</a>, tabs: "Dane archiwalne".
- IMERG (NASA, 2025).
- PDIR-Now: <a href="https://persiann.eng.uci.edu/CHRSdata/PDIRNow/PDIRNow1hourly/">https://persiann.eng.uci.edu/CHRSdata/PDIRNow/PDIRNow1hourly/</a>.
- ERA5 data (ECMWF, 2025).
- Other data used in this paper is available upon request, provided it is not restricted by its producer.

- Author contributions. AJ, JS, and KO developed the concept for the paper. MF carried out simulations
- with the WRF model. MS developed the software and carried out processing of various types of data.
- KO and AK developed the software and carried out the statistical calculations. AJ, JS, KO, MS, and AK
- conducted an analysis of the results. JS, AJ, and KO wrote the text of the paper. MS, AK, RP, and MF
- verified the text. JS and AJ made figures.

- Competing interests. The contact author has declared that none of the authors has any competing
- interests.

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
