# Peer review of "Can we reliably estimate precipitation with high resolution during"

_EGUsphere, 2025_

## Author Comment (AC1)

**Reviewer #1 (Nicolas Velasquez)**

The authors present a comprehensive comparison of multiple rainfall products against a collection of four manual rain gauges. The comparison helps to identify the quality of the sources potentially useful for early warning implementations. The authors focus their work on comparing point data with pixel data, which has been widely accepted by the community. However, point-based comparisons assume point observations as the ground truth, and they are, but for a limited area of space. On the other end, a comparison with more significance will integrate streamflow observations with rainfall accumulations at the watershed scale. At the end of the day, the rainfall totals and localization controls most of the watershed response during a flood event, not the rainfall that happened at a point where we were technically able to measure it.

> Clarification: not "against a collection of four manual rain gauges", but 112 gauges (see line 310 and Fig. 4). Four selected stations were only chosen to present more detailed analyses in Section 4.5 (Table 6 and Figure 10).

> As for the verification of precipitation based on catchment outflow: our aim was to assess the reliability of the precipitation available online by comparing it with the most reliable data, i.e. primarily from manual rain gauges.

Rainfall comparisons are always important for the community, and therefore, this work could be of significance. However, it needs to address some major issues. Following, I present my general and specific comments.

**General comments**

The abstract does not provide a description of the techniques used by the authors to make the comparisons. Also, it does not clarify why it is important to have a precise estimation of the rainfall. Finally, they don't relate their results to a significant outcome.

> We have changed the abstract:

>> "**Abstract**. A huge and dangerous flood occurred in September 2024 in the upper and middle Odra river basin, including mountainous areas in south-western Poland. The event provided an opportunity to investigate the feasibility of reliable estimation of high-resolution precipitation field, which is crucial for effective flood protection. Different measurement techniques were analysed: rain gauge data, weather radar-based, satellite-based, non-conventional (CML-based) and multi-source estimates. Apart from real-time and near real-time data, later available reanalyses based on satellite information (IMERG, PDIR-Now) and numerical mesoscale model simulations (ERA5, WRF) were also examined. Reference data used to verify the reliability of the different techniques for measurement and estimation of precipitation included observations from manual rain gauges and multi-source estimates from the RainGRS system developed at IMGW for daily and hourly accumulations, respectively. Statistical analyses and visual comparisons were carried out. Among the data available in real time the best results were found for rain gauge measurements, radar data adjusted to rain gauges, and RainGRS estimates. Fairly good reliability was achieved by non-conventional CML-based measurements. In terms of offline reanalyses, mesoscale model simulations also demonstrated reasonably good agreement with reference precipitation, while poorer results were obtained by all satellite-based estimates except the IMERG."

In section 2.1, the authors mention a collection of studies done before on flooding at the Odra River. However, they don't mention the main conclusions of these studies and how they relate to their current work. This must be addressed.

> We have completed section 2.1 with the following paragraph inserted at its end:

> "The above studies indicate that the upper Odra River basin is highly vulnerable to flooding caused by intense precipitation in the mountainous part of the basin. This is also influenced by the shape of the river network, which favours the cumulation of floods from individual tributaries. The flood risk there occurs almost annually during the summer."

The main comparison happens at four manual gauges located over the mountains in the south-west area of the watershed. Nevertheless, mountains tend to generate large disturbances over the rainfall patterns, probably biasing the results. Also, I am missing a figure in Section 2 presenting the rainfall accumulation of the event. Despite magnitude inaccuracies, a total accumulation figure will illustrate the area of interest for this event. It will also tell if the use of these gauges is worth it or not. You can make that figure by accumulating the rainfall from Figure 5.

> As we wrote above, the main reference data were not from 4, but from 112 manual rain gauges (see lines 310-311).

> In terms of rainfall accumulation of the event you are right: we have made a new figure in section 2.2 with the 4-day rain gauge accumulation from the whole flood period and have inserted it in line 191:

[Figure]

Figure 2: Field of precipitation accumulation during the flood of 13-16 September 2024 (four days) for the upper and middle Odra River basin in Poland, obtained from the RainGRS Clim reanalysis (GRS Clim).

It is troublesome to compare 1km or 0.5km rainfall products with point data. More in a mountainous region where convective rainfall was probably dominant. Point data represents a relatively insignificant portion of a 1km$^2$ area, and therefore, it may be highly biased. Instead, the watershed should be used to compare rainfall totals. Also, it is at the watershed scale when this comparison matters for flood forecasting and assessment purposes.

> Our aim was to assess the reliability of online precipitation by comparing it with the most reliable data, i.e. measurements by manual rain gauges. In the case of the floods described in our article, we mainly observed widespread precipitation, which generally creates smaller problems when comparing with grid data.

It is hard to conclude on the performance of a rainfall product based on the comparison of a single event. Conditions such as the season and the covered area are likely to induce a change in the performance. Additionally, the gauged records cover only a fraction of the watershed. The authors should consider expanding their analysis to more cases in the watershed.

> We completely agree with the Reviewer that the detailed comparative analysis of the different precipitation measurement techniques does not allow us to draw general conclusions about their

effectiveness. This would require an analysis of a really large number of events, at different seasons, in areas of different character and orography, with different types of precipitation, etc. Our aim was only to examine the effectiveness of high-resolution estimation of the precipitation field during an extreme precipitation event that happened in a specific basin. We believe it is worth showing how the techniques used worked in this specific situation.

Moreover, we do not have the possibility to carry out analogous analyses for other events. During previous major floods (2010, 1997), IMGW did not have such an extensive database of online measurements.

The English needs some editing.

We have revised the text in terms of language.

**Specific comments**

Lines 45-47: The authors talk about manual rain gauge measurements. At this point, they should be talking about automation. Manuals are fine for citizen science and community-based projects. However, these gauges have significant limitations in the temporal scale, especially when it comes to flood forecasting in mountainous regions.

The reviewer is of course right that the temporal resolution of manually measured data is low. However, they are still the most reliable precipitation measurements available, but given their limitations in terms of time scale, we used multi-source estimates based on data from automatic rain gauges, radar and satellites as an auxiliary reference for 1-h analyses (Tables 3 and 5). These data proved to be the most reliable when verifying daily accumulations based on manual rain gauges (Table 2).

Lines 48-50: There is an extensive literature on this topic. However, the authors don't mention it.

We have added literature references in line 51:

Hohmann, C., Kirchengast, G., O, S., Rieger, W., and Foelsche, U.: Small catchment runoff sensitivity to station density and spatial interpolation: hydrological modeling of heavy rainfall using a dense rain gauge network, Water, 13, 1381, doi:10.3390/w13101381, 2021.

Loritz, R., Hrachowitz, M., Neuper, M., and Zehe, E.: The role and value of distributed precipitation data in hydrological models, Hydrology and Earth System Sciences, 25, 147–167, doi:10.5194/hess-25-147-2021, 2021.

Lines 67-73: Same, I am missing literature on this matter here.

We have added a link in line 72 to the already existing reference: Ośródka and Szturc (2022) and this new reference:

Méri, L., Gaál, L., Bartok, J., Gažák, M., Gera, M., Jurašek, M., and Kelemen, M.: Improved radar composites and enhanced value of meteorological radar data using different quality indices, Sustainability, 13, 5285, https://doi.org/10.3390/su13095285, 2021.

Lines 93-104: Better referencing. However, the authors failed to mention some of the most important operational products on this matter, such as the Multi-Radar Multi-Sensor (MRMS) product provided by NOAA.

We have added a reference to MRMS in line 104:

> "NOAA provides the Multi-Radar Multi-Sensor (MRMS) quantitative precipitation estimation product automatically generated through integration of data from radar networks, surface and satellite observations, numerical weather prediction (NWP) models, and climatology (Zhang et al., 2016)."

> Zhang, J., Howard, K., Langston, C., Kaney, B., Qi, Y., Tang, L., Grams, H., Wang, Y., Cocks, S., Martinaitis, S., Arthur, A., Cooper, K., Brogden, J., and Kitzmiller, D.: Multi-Radar Multi-Sensor (MRMS) quantitative precipitation estimation: Initial operating capabilities. Bulletin of the American Meteorological Society, 97, 621–638. https://doi.org/10.1175/BAMS-D-14-00174.1, 2016.

Lines 127-130: It is still not clear to me how a manual rain gauge is able to measure the details of a storm. For my understanding, a manual rain gauge refers to a gauge that requires a human to read it (like an accumulation bucket). Maybe I am wrong, but I know that this can also confuse other people who work in this area.

> The manual rain gauge obviously does not measure the details of the temporal precipitation distribution during a storm, but the daily accumulation are the most accurate of all techniques, so we used the GAU Manual data to verify daily accumulations. Of course, these measurements do not allow analysis of rainfall on shorter time scales, so we used multi-source GRS estimates to verify hourly accumulations.

Lines 137-139: From my experience, precipitation fields of 1km every 10 minutes or every hour can still be too coarse for flash flood forecasting in mountainous regions. In many of these rivers, the response time is in the order of minutes, and the variability of the convective systems is in the order of $m^2$. The authors should contrast their statement with the literature on floods in mountainous regions. In your case, we are talking of a large watershed (44,000 $km^2$) and therefore, the scale and the time frame are ok. However, you should state that in your introduction; otherwise, readers may get confused.

> We have revised the paragraph in lines 137-142:

> > "The main objective of this work is to examine the real possibilities of precise estimation of a precipitation field with a high spatial resolution of about 1 km and a high temporal resolution of at least 10 min, or one hour during intense precipitation events that cause floods in upper Odra River catchment area in September 2024. All available real-time and offline measurements and estimates were verified to determine their applicability and to quantify their reliability."

Line 167: The reference has an issue.

> We have corrected the bibliographic description in References in lines 934-938:

> > "Ligenza, P., Tokarczyk, T. and Adynkiewicz-Piragas, M. (Eds.): Przebieg i skutki wybranych powodzi w dorzeczu Odry od XIX wieku do czasów współczesnych (The course and impacts of selected floods in the Oder river basin from the XIXth century to the present day), Instytut Meteorologii i Gospodarki Wodnej − Państwowy Instytut Badawczy, Warszawa, 2021, 132 pp., ISBN: 978-83-64979-45-3, 2021 (in Polish)."

> and a reference in the text to this item in line 167: „*Przebieg...*, 2021" to: „Ligenza et al., 2021".

Line 176: "and other works by this author" change that and add the proper citations. They could be important for some readers.

> We have changed the citation „Kundzewicz, 2014" into:

> > „e.g. Kundzewicz, 2014"

Lines 180-181: revise the phrase that is between these lines. It is hard to understand.

We have changed the sequence into:

"Research suggests that climate change affects the frequency and severity of floods, leading to an increased risk of flooding (e.g. Kundzewicz et al., 2023)."

Line 187: Change "per" for "in"

We have changed.

Lines 192-193: Add the reference after the 400 mm.

We have removed the reference to Kimutai et al. (2024) because we cited it not because of the amount of rainfall, but because of the use of a 4-day total.

Table 1: The authors are mixing units of km and degrees. They should transform the degrees according to the watershed projection. They only provide the transformation for ERA5. It is hard to make a fair comparison if they don't use the same units. You show this information in line 261. Present it on your table.

Thanks for this comment! We have supplemented the information in Table 1 with the pixel sizes in km:

- rows SAT and H61B: "Roughly 5-6 km for Poland" → "Roughly 3.5 km x 6.0 km*"
- row IMERG: "0.1º x 0.1º" → "Roughly 7 km x 11 km* (0.1º x 0.1º)"
- row PDIR-Now: "0.04º x 0.04º" → "Roughly 2.8 km x 4.5 km* (0.04º x 0.04º)"
- row ERA5: "0.25º x 0.25º" → "Roughly 18 km x 28 km* (0.25º x 0.25º)"
- Footnote: "* Roughly 18 km x 26 km." → "* In the area of the study basin."

Figure 2: Should present the radar beam radius (50, 100, or 150 km?). This is also important when contrasting rainfall estimates from radar data. Also, if possible, the figure should present the blind spots for the radars due to the mountains.

We have added circles with a radius of 150 km around the radars to Fig. 2, because our experience with radar-based precipitation shows that up to such a distance the data retain the highest reliability, if, of course, there are no terrain obstacles in the path of the beam. There are mountains in the south-eastern part of the study catchment, but these are surrounded by as many as eight radars. Thus, there are no places in the study area where the lowest radar beam is too high above ground. It can therefore be assumed that there are no places in the area for which high quality radar data would not be available.

Figures 3 and 4 look redundant. Their information can be condensed in Figure 2.

Following the Reviewer's suggestion, we have combined the former Figs. 3 and 4: in Fig. 3 there are all measurements in near real time: automatic (telemetric) rain gauges, weather radars, and CMLs, and in Fig. 4 there are only manual gauges (we have only slightly changed the station symbols).

[Figure]

Figure 3: Locations of measurement stations in the upper and middle Odra River basin: telemetric rain gauges (blue dots), weather radars (brown triangles) with 150-km range (brown circles), commercial microwave links (black lines), and four manual rain gauges selected for more detailed analysis (larger blue dots).

[Figure]

Figure 4: Locations of manual rain gauges (blue circles) and four ones selected for more detailed analysis (larger blue dots) in the upper and middle Odra River basin.

Lines 504-505: If I understand well, the comparison corresponds to days where only 50mm or more was observed by the manual gauges. If this is correct, explain it better.

We have changed this sentence to:

"The results of the statistical analysis based on daily accumulations from manual rain gauge measurements (GAU manual) for days with recorded rainfall of 50 mm or more are presented in Table 4."

Lines 530-532: Why 5mm? why 200 pixels?

These values were chosen empirically in such a way as not to take 1-hour totals with low precipitation into the statistics, as they introduce randomness into the results, especially as regards the CC.

Lines 534-536: The argument lacks scientific support.

We have removed the last sentence from this paragraph (lines 534-536) and added a new one at line 532:

„Thresholds of 5-mm for hourly accumulations and 200 pixels for the area where such precipitation occurred (approximately 0.5% of the entire catchment) were introduced to exclude data with low precipitation from the statistics."

Tables 2 to 5: These tables look repetitive and hard to read. A reader probably won't take an important message from them. Instead, a collection of figures (one next to the other) presenting the comparisons done under the different conditions should be more illustrative. Ideas for such a collection include, but are not limited to:
− Scatter plots comparing the correlation vs the RMSE for each case. One case next to the other.
− Scatter plots comparing the metric for the daily case (x axis) and the other cases (y axis).

We made graphs of CC depending on RMSE and inserted them as Figs. 8, 9, 10, and 12. They illustrate the results from Tables 2, 3, 4 and 5, respectively.

[Figure]

Figure 8: Scatter plot comparing CC vs RMSE for each measurement and estimation technique for daily precipitation accumulations from 13-16 September 2024, against data from manual rain gauges (GAU Manual) as reference.

[Figure]

Figure 9: Scatter plot comparing CC vs RMSE for each measurement and estimation technique for hourly precipitation accumulations from 13-16 September 2024, against the RainGRS estimates (GRS) as reference.

[Figure]

Figure 10: Scatter plot comparing CC vs RMSE for each measurement and estimation technique for daily precipitation accumulations from 13-16 September 2024 against data from manual rain gauges (GAU Manual) as a reference with a threshold of 50 mm.

[Figure]

Figure 12: Scatter plot comparing CC vs RMSE for each measurement and estimation technique for hourly precipitation accumulations from 13-16 September 2024 against the RainGRS estimates (GRS) as a reference with a threshold of 5 mm.

Table 6: More boring tables that most likely nobody will read. This large collection of numbers in the middle of a paper is not telling a story. Instead, it makes the manuscript look like a consulting report. Consider other ways of showing the differences between the hietograms. The table can go to an appendix section.

We have moved Table 6 to Appendix A.

Lines 627-628: This is an expected outcome.

Yes, this is a rather expected observation, but we think it is worth emphasising here.

Section 4.6: Can be significantly improved if the authors include a schematic summarizing their results.

We have introduced diagrams 8, 9, 10, and 12, which intuitively illustrate the results obtained. We therefore believe that it is no longer necessary to do an additional schematic summarizing of the results.

---

## Author Comment (AC2)

**Reviewer #3**

**General comments**

The paper was well written, containing a good introduction to readers also outside meteorological and hydrological community. Also the topic – uncertainties in predicting intensive flooding – is very relevant in modern society still sensitive to weather conditions. The geographical scope of this study was limited to Poland, but it can be expected that the results are applicable in Southern and Northern climates as well, especially in mountainous areas.

**Specific comments**

The experiments section involved verification of products that depend on inputs that are used as ground truth in comparisons. This was clearly pointed out throughout (L377, L426, L485, L548, L556, L582, L621) the article which is of course appreciated. Nevertheless, interpreting verification is problematic in these cases. On one hand, for example, one could easily think of multi-input algorithm design where output if asymptotically forced to match point measurements used also as reference – yielding zero error. Or if the values do vary, it would be good to know motivations in design (for example, avoiding overfitting). Nevertheless, such evaluations have little or no information in my opinion. On the other hand, experts in this area do know the challenge of evaluating input sources (measurement technologies), none of which is perfect (L95) . It is operationally tempting – if not inevitable – to combine inputs of various type (L123). But as to verification of performance, one should try to use reference data as independent as possible. Would it be easy to use some kind of cross-validation, dropping some ground observations (in turn) from input, and to verify the results against those? This could yet require more computational effort.

> We thank the Reviewer for these comments! Of course, we discussed a lot in the authors' team about this topic, and finally:
>
> – We agree that "one should try to use reference data as independent as possible". Therefore, if we use data that is not completely independent, we always include the annotation "(dependent)" in the results tables.
> – We have removed the GRS Clim fields from the verification of the 24-h accumulation because the final step in the development of the product is to adjust it to the GAU Manual, so later verification with the same data cannot give fully reliable results. All other data after removing GRS Clim, are independent of the GAU Manual data.
> – Because it is impossible to use the daily GAU Manual accumulations to verify 1-h accumulations, we chose the 1-h multi-source GRS estimates as reference data, which, as shown in Table 2 shows, are the best estimators of 24-h totals.
> – Thus, in 1-h verifications with GRS as reference we have focused on the reliability of other estimates on which GRS does not depend in any way, e.g. satellite precipitation, from NWP models, and unconventional techniques.

When evaluating prediction based on commercial microwave links (CML), a natural explanation (L423,L584-585) of deviations is distance from reference measurements (GAU manual). Could it be useful to study the effect of distance by measuring correlation inside reference data itself? Then input data from a separate system (like CML) could be then compared against such modelled, "theoretical" maximum – providing estimates of measurement uncertainty at least in the vicinity of the links. (This is more a suggestion for further work, not for this study and this could be of more interest for CML application developers.)

*Thank you very much for these suggestions! We are currently working hard on the calibration, quality control, and operational implementation of CML data, which proves to be not so easy. We will definitely use this suggestion of autocorrelation analysis of GAU Manual data.*

The article reports errors in predictions using input from radar and especially, satellites. Many potential error sources (L58, L59; L69) are well-known – like measurement geometry or uncertainly of water phase (in both radar and satellite measurements). It would be interested to read the author's views on which of the error sources have been critical in this study. Systematic analysis could be certainly outside the scope of this article, but perhaps just visual inspection could be used as a basis for discussion on error sources.

*We have supplemented section 4.2 with our comments on this topic:*

*to line 416:*

*"The radar network in the analysed flood area is relatively dense, but due to signal blocking by mountains, precipitation shadows appear in some places, which result in an underestimation of precipitation. This is particularly evident in the Kłodzko Valley which is surrounded by relatively high mountains and is one of the places most prone to catastrophic flooding."*

*to line 420:*

*„The reliability of precipitation estimates based on satellite data is low, especially when they are generated from infrared channel data and are not supported by other, preferably microwave data (from radars). This mainly affects SAT estimates, but also others. It should be noted that during the analysed flood, data from visible channels was only available for about 1/3 of the time, due to the fact that for the measurements to be reliable, the sun must be sufficiently high above the horizon (above 20 degrees). Furthermore, the spatial resolution of these data is generally insufficient."*

Focusing separately in cases of intensive rainfall (Sec 4.4.) is well motivated. When thresholding the cases with reference (L505, L531), negative bias is reported for all the methods (Table 4), also highlighting it for radar in text (L510, L546). Especially in verifying GAU against GAU Manual, I think that the reported underestimation (L573, L574) is a direct consequence of the applied thresholding! Consider two measurement devices of similar climatology some kilometres apart and long-term statistics of (convective) rainfall: measured rainfall is then similarly distributed over the mean value. But if studied cases are limited by thresholding data on ONE measurement location/device, the other still includes also the lower values, pushing its bias down! (Consider throwing two dice, comparing averages of each, but limiting the studied cases by thresholding the first die.) I guess also with radar, similar effect can be observed when limiting cases by thresholding the reference value. (Radar's bias could be basically zero, but "random noise" ie. positive and negative deviation around mean is now caused by non-uniform vertical profiles of precipitation and advection, for example.) If you agree with me in this, I suggest you somehow address and elaborate this in text and/or presented experiments.

*We are aware of the limitations of a methodology of excluding precipitation accumulations that are below a certain threshold from the statistical analysis. We agree that its application requires commentary and an indication of the consequences of these limitations. Thus, in place of the sentence on lines 506-507 in Section 4.4, we have inserted the following fragment:*

*"As expected, the results are noticeably worse when compared to those obtained without a limitation on precipitation magnitude (see Table 2). This is particularly evident in terms of bias, which indicates an increase in underestimation. However, a negative bias was observed for all the estimation techniques analysed, even without thresholding. This suggests a real underestimation of intense precipitation by these methods, rather than simply a result of data selection."*

**Technical comments**

RainGRS is mentioned several times before explained or referenced. It is also unclear, what is "RainGRS (GRS)" compared to plain "RainGRS".

> We have reordered this. In the abstract where the name "RainGRS" first appeared we have changed the sentence in lines 18-21 to:
>
> > "Reference data used to verify the reliability of the different techniques for measurement and estimation of precipitation included observations from manual rain gauges and multi-source estimates from the RainGRS system developed at IMGW for daily and hourly accumulations, respectively."
>
> "GRS" are multi-source estimates from RainGRS system – see Table 1.

Long URLs embedded in the text (L328, L337, L345, L353) reduce readability a bit (). If the publisher's style guide supports it, could they be in the references?

> We have included links to references:
>
> line 328:
> > NASA. GPM IMERG Final Precipitation L3 Half Hourly 0.1 degree x 0.1 degree V07 (GPM_3IMERGHH): https://disc.gsfc.nasa.gov/datasets/GPM_3IMERGHH_07/summary?keywords=%22IMERG%20final%22, last access: 29 July 2025.
> line 337:
> > CHRS, https://chrsdata.eng.uci.edu/, last access: 29 July 2025.
> line 345:
> > ECMWF. ERA5 hourly data on single levels from 1940 to present: https://cds.climate.copernicus.eu/datasets/reanalysis-era5-single-levels, last access: 29 July 2025.
> line 349:
> > NCAR. Weather Research and Forecasting model WRF: https://ncar.ucar.edu/what-we-offer/models/weather-research-and-forecasting-model-wrf, last access: 29 July 2025.
> line 353:
> > DWD. ICON (Icosahedral Nonhydrostatic) Model: https://www.dwd.de/EN/research/weatherforecasting/num_modelling/01_num_weather_prediction_modells/icon_description.html, last access: 29 July 2025.

A minor detail: place names seem to have mixed style; English names should be preferred if they exist. (According to Wikipedia, Oder seems to be the established English name for the river Odra (PL). Also Sudetes (EN) and Sudety (PL) appear, but understandably the smaller the locations/regions, less English names exist! Anyway, I leave it to the authors to decide the naming policy.)

> We have standardised the nomenclature throughout the article. We have left geographical names in Polish, as they are more commonly used in the literature.

---

## Author Comment (AC3)

**Reviewer #2**

**General**

The authors analyse different precipitation data based on measurements from different sources as well as numerical weather prediction (NWP) model data. The measurement data include rain gauges, weather radar, satellite data and commercial microwave link (CML) data or products based on these data. The precipitation data are analysed for an extreme precipitation event on the Odra river catchment which occurred during four days in September 2024.

The paper is presenting the different results in a clear manner and discusses the advantages and disadvantages of the different measurement / modelling principles in detail.

In particular, limitations for satellite and NWP products can be demonstrated, while rain gauge and radar based products perform the best.

**Detailed discussion items**

Line 124 ff: Reference for the analysis:

The authors discuss the relevance for a reference data set which is independent from the other data sets. Unfortunately, they later include precipitation products that are not independent from the reference data set.

- While for daily values, the manual gauges are selected as reference - a choice which is the best possible from the available data -, the post processed GRS Clim product which is adjusted based on these reference station data enters into the investigated methods. This should be avoided because it deviates the attention of the reader from the relevant data to be compared.

> We have agreed with the Reviewer's arguments and removed the GRS Clim estimates from all analyses. In particular, we have revised the paragraph 191-198 in section 2.2:

>> "At many locations, the daily precipitation accumulation in this period exceeded 200 mm, and its territorial range covered mainly the Eastern Sudetes. Four-day precipitation accumulation reached values above 400 mm, with the highest in the Jeseníky and Śnieżnik Mountains. They might have exceeded even 550 mm, as indicated by reanalyses RainGRS Clim (Jurczyk et al., 2023) based on estimates from the Rain GRS system adjusted to observations from manual rain gauges (Fig. 2). Apart from intense, widespread precipitation, numerous thunderstorms and several associated tornadoes were recorded during these days. On 16 September, rainfall began to diminish; mainly light to moderate precipitation was observed, and in the following days, the weather in Poland was influenced by a high-pressure system, with the advection of warm and dry air of continental origin."

- Concerning hourly values, the choice of an independent reference of high quality is not solvable with the existing precipitation products. Therefore, the choice made by the authors is understandable, in particular since they are pinpointing this methodological weakness.

> OK.

Line 379ff: Selected metrics

When comparing gridded data to point data at the ground, location uncertainties may arise because rainfall observed at a certain height (or from space) does not necessarily fall down at the point which is considered by overlaying grids and points. Furthermore, uncertainties of comparing a grid area average to a point value occur in particular in heavy rain - in the order of 20% have been observed (Schellart et al., 2017). Can you please add a short section on how you take into account such uncertainties or, alternatively, which range of values has to be considered reliable?

We agree with the Reviewer that the problems associated with uncertainty in comparing point and grid data are important, although in our view it is challenging, especially when analysing such short accumulations. So we have added the relevant sentences to line 136 (in sect. 1.2.4):

"Furthermore, comparing the average precipitation over a grid area to a specific point value introduces some uncertainty, particularly during heavy rain (Ensor and Robeson, 2008). An analysis of findings by Sun et al. (2018), Herrera et al. (2019), and others shows that, due to the high spatial variability of precipitation, it is not possible to establish a single universal error value when comparing point and grid data. The level of the uncertainty varies depending on the nature of the precipitation. For widespread (large-scale) precipitation, the uncertainty typically ranges from about 10% to 15%. However, for intense, convective extreme precipitation, this uncertainty can rise to approximately 15% to 25% (Schellart et al., 2017; Henn et al., 2018; Tarek et al., 2021). Special care should be taken when analysing local precipitation maxima using gridded data, as noted by Sun et al. (2018) and others, who point out that these data may smooth out extreme values compared to point measurements."

New references:

Ensor, L. A. and Robeson, S. M.: Statistical Characteristics of Daily Precipitation: Comparisons of Gridded and Point Datasets. Journal of Applied Meteorology and Climatology, 47, 2468–2476, https://doi.org/10.1175/2008JAMC1757.1, 2008.

Henn, B., Newman, A. J., Livneh, B., Daly, C., and Lundquist, J. D.: An assessment of differences in gridded precipitation datasets in complex terrain. Journal of Hydrology, 556, 1205-1219, https://doi.org/10.1016/j.jhydrol.2017.03.008, 2018.

Sun, Q., Miao, C., Duan, Q., Ashouri, H., Sorooshian, S., and Hsu, K.-L: A review of global precipitation data sets: Data sources, estimation, and intercomparisons. Reviews of Geophysics, 56, 79-107, https://doi.org/10.1002/2017RG000574, 2018.

Tarek, M., Brissette, F., and Arsenault, R.: Uncertainty of gridded precipitation and temperature reference datasets in climate change impact studies, Hydrology and Earth System Sciences, 25, 3331–3350, https://doi.org/10.5194/hess-25-3331-2021, 2021.

Herrera, S., Kotlarski, S., Soares, P. M. M., Cardoso, R. M., Jaczewski, A., Gutiérrez, J. M., and Maraun, D.: Uncertainty in gridded precipitation products: Influence of station density, interpolation method and grid resolution. International Journal of Climatology, 39, 3717–3729, https://doi.org/10.1002/joc.5878, 2019.

Do the selected metrics show well the effects that you are most interested in, i.e. the best estimate for extreme intensities and also for the cumulated sums? Squared error indices tend to heavily penalize individual outliers which may be one effect that you are after, but please discuss this aspect.

In line 390 we have added:

„The RMSE is particularly sensitive to outliers as squaring the errors emphasizes larger deviations."

In line 391 we have added:

"as it relates the deviations to the spread of the reference values around their mean"

Line 553: Do you want to say that interpolated gauges are more reliable than adjusted radar data? Then you contradict yourself because earlier you said that interpolated station data are underestimating the true values.

> The sentence in lines 552-555 is incorrect. We have corrected it to:

> > "The conclusion from this analysis is that the estimation of extremely high precipitation fields with very high spatial (1 km) and temporal (1 hour) resolution is mainly based on weather radar observations, but these must first be adjusted to the rain gauge data. Rain gauges can also produce reliable estimates, but under the condition that a sufficiently dense network of such gauges is available."

**Formal aspects**

Line 14: please replace "... 200 mm daily" by "... 200 mm on one day at one rain gauge location"

> We have significantly rebuilt the abstract, and as a consequence this fragment is no longer included.

Line 71ff: please give the explanation for each abbreviation before using it (RLAN, GPM, NOAA, MetOp, GAU, etc.)!

> We have added abbreviation expansions:
> RLAN in line 71 (Radio Local Area Network)
> GPM in line 80 (Global Precipitation Measurement)
> NOAA in line 80 (National Oceanic and Atmospheric Administration)

Line 191: please start the sentence with "In many locations, the daily precipitation ..." - the values in the tables given later suggest that this formulation is more precise.

> Changed.

Line 366: please rephrase to something like "... which we consider to be the most reliable values."

> Changed.

Line 479: according to Table 2, the Bias is -3.8 mm (not -3.6 mm) - one of the two should be corrected...

> Corrected to "-3.8" in the text.

**Technical points**

Lines 274-278: This explanation should be clarified - which other approaches were tested before selecting the final method and by which means is it different to the others? Please also refer to the results from the COST OpenSense Action!

> We have removed the sentence in lines 276-278, because it does not explain anything, and the methods are described in the in the paper by Pasierb et al.

> We have added a reference to the paper here: Olsson et al. (2025) in line 274.

Line 309: You are writing "closest to reality" - however, this is correct for one point and is of limited value for areas. Please emphasize it here again, although you mentioned this earlier already.

Corrected to:

"The data from the manual rain gauges are the closest to reality at their locations, and therefore were selected as the point reference for the 2024 flood."

Lines 329 - 331:

- "...satellite data as a base line and intercalibrates." What is intercalibrated here?
- "... other observations with international satellite constellation ..." Please note that GPM as the Global Precipitation Measurement mission is heavily based on satellite-based weather radars. The chosen formulation suggests that GPM does not include radar and these data need to be retrieved from other sources

We have corrected the sentences to (lines 329-332):

"This product is calibrated with Global Precipitation Measurement (GPM Core Observatory) satellite data, which is based on microwave imager and the dual-frequency precipitation radar, and uses it as a baseline. It is combined with other observations from national or international satellite constellations equipped with weather radars and passive microwave and infrared sensors, as well as with rain gauge data (Huffman et al., 2020; Bogerd et al., 2021)."

Lines 347-348: How can you analyse short lived phenomena if your resolution is not sufficient for convective cells? Please explain!

We have changes the sentences to:

Such data allows for an overall analysis of rainfall offline. However, it is impossible to use these reanalyses when knowledge of the course of convective phenomena at the microscale is needed, i.e. with a spatial resolution of 1 km or less.

Lines 365 to 378: I understand it correctly that the daily analysis relies on 112 data points (= all manual stations in the area) and the hourly analysis on statistics calculated from 44218 pixels? If so, please add the numbers here for a better understanding!

We have completed the sentence in lines 366-367:

"These measurements are point wise, so verification of individual precipitation fields was performed only at the locations of these stations (112 ones)."

…and in lines 374-375:

"As measurements from manual rain gauges are not available at such a short time step, the RainGRS (GRS) fields (44,218 pixels within the basin) were used as a benchmark for the verification."

Line 411-412: What is the influence of data from the Czech territory? I do not understand.

The GRS multi-source precipitation field (generated by the RainRGS system) is created from, among others, the precipitation field resulting from spatial interpolation of rain gauge measurements. The amount of precipitation in each pixel is influenced by the nearest rain gauges. Near the border with the Czech Republic, rain gauges located on the other side of the border also contribute to this interpolation. Hence this influence, which is only visible close to this border.

Lines 413: RAD data product: please eliminate the discussion of unadjusted radar data - else readers may think that they can work with such data. Merely, a warning would be adequate to never use unadjusted radar data for any quantitative purpose, maybe with a reference to the WMO Operational Weather Radar Best Practice Guidance (WMO document no. 1257 - https://library.wmo.int/records/item/68834-guide-to-operational-weather-radar-best-practices?offset=5).

The Reviewer is right - thank you for this comment. Indeed, the paragraph as it is now may lead to confusing conclusions. We have changed it as follows:

"In the case of radar-derived fields (RAD and RAD Adj), the precipitation pattern is also well represented, but the estimate based solely on radar observations (RAD) underestimates values. Therefore, unadjusted radar data should not be used, especially for quantitative precipitation estimates (WMO-No. 1257, 2025). Radar data after adjustment with rain gauge measurements (RAD Adj) demonstrates good agreement concerning precipitation values."

We have added this item in References:

WMO-No. 1257: Guide to Operational Weather Radar Best Practices. Volume VI: Weather Radar Data Processing, World Meteorological Organization, Geneva, 156 pp., https://library.wmo.int/records/item/69563-guide-to-operational-weather-radar-best-practices (last access: 16 July 2025), 2025.

References:

- Schellart ANA, Wang L & Onof C (2017) High resolution rainfall measurement and analysis in a small urban catchment. 9th International Workshop on Precipitation in Urban Areas: Urban Challenges in Rainfall Analysis, UrbanRain 2012 (pp 115-120)

We have added this paper - we were not aware of it before. Thank you for pointing it out!

Schellart, A.N.A., Wang, L, and Onof, C.: High resolution rainfall measurement and analysis in a small urban catchment. 9th International Workshop on Precipitation in Urban Areas: Urban Challenges in Rainfall Analysis, UrbanRain 2012, ETH Zurich, 115-120, ISBN 978-390603121-7, 2017.